# Antibacterial Drug Residues in Small Ruminant Edible Tissues and Milk: A Literature Review of Commonly Used Medications in Small Ruminants

**DOI:** 10.3390/ani12192607

**Published:** 2022-09-28

**Authors:** Emily D. Richards, Krysta L. Martin, Catherine E. Donnell, Maaike O. Clapham, Lisa A. Tell

**Affiliations:** 1Food Animal Residue Avoidance and Depletion Program and Department of Medicine and Epidemiology, School of Veterinary Medicine, University of California-Davis, Davis, CA 95616, USA; 2School of Pharmacy, University of North Carolina-Chapel Hill, Chapel Hill, NC 27514, USA

**Keywords:** small ruminant, sheep, goat, milk, edible tissue, antibiotic drug residue

## Abstract

**Simple Summary:**

This review is a summary of published studies that contain drug residue depletion data for edible tissues and milk following treatment of sheep and goats. The information is separated by antibiotic class for ease of comparison between studies. This summary is useful for understanding medication residue depletion following extra-label drug use and can be used to help estimate withdrawal intervals in order to help protect the human food chain.

**Abstract:**

This review provides a summary of extracted data from the published literature that contains drug residue depletion data for edible tissues and milk following treatment of sheep and goats. Out of 20,234 records obtained during the initial search, data from 177 records were included in this review. The data is separated by antibiotic class for ease of comparison between studies. Extracted data includes the active ingredient, dosing information, animal health status, analytical method and limits of detection, tolerance and maximum residue limit information, and time frames relative to residue absence or detection. This information is useful for understanding drug residue depletion profiles following extra-label use and for estimating withdrawal intervals, in order to protect the human food chain.

## 1. Introduction

Drinking water and availability of food for both humans and animals are affected by climate change that lowers rainfall and an increasing world population, especially in semi-arid climates [1]. Small ruminants present a unique opportunity for developing nations, specifically in developing nations that are in semi-arid climates, due to their multi-purpose use (meat, milk and fibers), lower production cost compared to large ruminants, and tolerance to low rainfall and hot climates [1].

According to data from the Food and Agriculture Organization of the United Nations (FAO), the number of sheep and goats worldwide has increased from approximately 1.4 billion head combined (1 billion sheep, ~400 million goats) in 1961 to approximately 2.3 billion head combined (~1.2 billion sheep, ~1.1 billion goats) in 2019 [2]. Between 2014 and 2019, the largest producers of sheep meat worldwide were China, Australia, New Zealand, Turkey and Algeria, whereas during this same time period, the largest producers of goat meat worldwide were China, India, Pakistan, Nigeria and Bangladesh.

In the United States, sheep and goats are considered minor species by the Food and Drug Administration (FDA) [3]; however, sheep are considered major species while goats are considered minor species by the European Medicines Agency (EMA) Committee for Medicinal Products for Veterinary Use [4]. In the United States, there is a “severe shortage of approved new animal drugs for use in minor species” [5].

The Food Animal Residue Avoidance and Depletion Program (FARAD) is a United States Department of Agriculture (USDA)-funded program with a mission to provide veterinary practitioners with scientifically based withdrawal interval recommendations following extra-label drug use or chemical/pesticide contamination in food-producing species. FARAD call submission data for small ruminants indicates a steady increase in the number of withdrawal interval request submissions from 2015 to 2019, with a steep increase in the number of submissions in 2020 (2015 = 435 submissions for sheep, 223 for goats; 2019 = 343 submissions for sheep, 710 for goats; 2020 = 595 submissions for sheep, 1401 for goats). The most commonly requested drug categories include antibiotics, anthelmintics, and non-steroidal anti-inflammatory drugs (NSAIDs). This data reflects the increasing numbers of backyard or hobby-farm environments, where the food-products are consumed by the family keeping the sheep or goats. Given the limited FDA-approved medications for use in sheep or goats, drugs are commonly prescribed in an extra-label manner which is legalized by the Animal Medicinal Drug Use Clarification Act of 1994 (AMDUCA) [6].

Given the importance of sheep and goats as commodity groups worldwide, the purpose of this review is to summarize research studies investigating antibiotic medication use in small ruminants with respect to the potential for drug residues to be present in small ruminant meat and milk products. Due to the large volume of published literature in small ruminants, this review only includes antibacterial medications; however, a second review will be completed incorporating anthelmintics and other medication classes not included here. It is important to note that residue depletion times referenced in the text are based on data from scientific studies. Normal industry practice to withdraw feed 8 to 12 h prior to processing the animals in order to minimize fecal contamination [7] may not have occurred in scientific research studies examining a zero day withdrawal. In addition, the residue depletion times listed in this manuscript are dependent on the sensitivity of the analytical method utilized in the study. Summaries of drug residue studies, drug approvals, tolerances (in the United States), and maximum residue limits (MRLs; in the European Union) have been provided in the tables for the reader’s convenience. If available, FDA-approved medications for use in sheep and goats should be utilized according to directions and labeled withdrawal times adhered to in order to guarantee human food safety.

## 2. Materials and Methods

### 2.1. Search Strategy

A systematic literature search was conducted using various databases and compared to publications included FARAD Program’s literature database. The aim of the search was to collect milk and edible tissue residue data for antibiotics that had been administered to small ruminants. Published literature between 1926 and 2021 was searched using PubMed, Cab Direct, Scopus, and Web of Science. Search terms and key words included: “sheep”, “goat(s)”, “small ruminants”, “caprine”, “ovine”, “drug absorption”, “clearance”, “drug residue(s)”, “pharmacokinetics”, “metabolic clearance rate”, “intestinal absorption”, “bioavailability”, “biological availability”, “metabolism”.

### 2.2. Screening Results

For systematic screening, search results were imported into the Covidence online platform (Covidence Systematic Review Software, Veritas Health Innovation, Melbourne, Australia) and duplicate results were removed by the Covidence software. Initially, the 20,234 “Titles and Abstracts” were screened by one reviewer (EDR or CED) for relevancy and categorized as ‘yes’, ‘no’, or ‘maybe’ using predetermined inclusion and exclusion criteria. The category of ‘maybe’ was used for trials that did not explicitly state the inclusion or exclusion criteria in the abstract and thus required further review of the full text. Inclusion criteria were as follows: in vivo sheep or goat drug trial; drug or metabolite concentration data and time point in tissue and/or milk; drug dose, route of administration, and dosing frequency stated. Exclusion criteria were as follows: any animal not a sheep or goat; in vitro study; concentration or residue data for non-drug substances (pesticide, toxin, vitamins) or drugs of abuse; drug plasma or serum concentrations only; dose of drug, route of administration, and dosing frequency missing. After initial screening for exclusion criteria, 1769 ‘yes’ and ‘maybe’ results moved to a ‘Full Text’ screening by one reviewer (EDR, KLM, or CED). These records were further excluded or included based on the above criteria and a reason was assigned. Records were excluded due to: not being a study (e.g., review, short communication, corrigendum; *n* = 128), not being able to verify text (e.g., full text not available from lenders worldwide, abstract only from proceedings, text unable to be translated; *n* = 141), being the wrong patient population/study design (e.g., not in live animals, in live animals other than small ruminants, etc.; *n* = 84), chemical product of study was a non-drug substance (*n* = 10), matrices under study did not consist of tissues or milk (*n* = 1076), and lack of specific concentration versus time presented in the paper (*n* = 60). A total of 270 records met the complete inclusion criteria. Figure 1 displays a flowchart representation of the screening process completed in this literature search.

For comparison, the FARAD database returned 832 records for both sheep and goats; however, 78 records were removed from the review due to incorrect matrices (i.e., plasma or serum data only). Ultimately, only 177 records met the complete inclusion criteria.

## 3. Data Extraction and Presentation (Antibiotic Drug Classes, Residue Detection, and Analytical Methods)

The published literature presenting tissue and milk residue data for antibiotics used in sheep and goats is presented in the Tables below and is categorized by antibiotic class. Tolerances or maximum residue limits are presented for FDA-approvals and EMA-approvals, respectively. The basic analytical method is described, with a focus on the limit of detection and limit of quantitation, alongside the dosing regimen for each study. Animal health status and additional information are also included, since variations in health- or lactation-status may affect drug residue depletion. Finally, two columns are included to indicate when residues were last detected. The column titled ‘Last sampling time point for which residues WERE detected (post-last treatment)’ refers the last sampling point when residues were detected based on the study sampling protocol. This is in contrast to the column titled ‘Sampling time point when NO residues were detected (post-last treatment)’ which refers to the last sampling point when residues were *not* detected based on the study sampling protocol. Instances where a greater than symbol (“>”) is utilized refers to situations where residues were still detected at the last sampling time point of the study protocol.

Data for the summarized studies includes analytical methods since it is important to consider how those methods impact the sensitivity of drug residue detection and how the analytical limits of detection compare to tolerances or MRLs. Newer analytical methods can detect drug residues at lower concentrations than historical microbiological bioassays or colorimetric testing, resulting in a greater number of days with detectable drug residues. In contrast, studies using less sensitive methods, having higher limits of detection, may have found shorter periods with detectable drug residues upon withdrawal of the drug. Readers are cautioned to keep the sensitivity of the analytical methods in mind when evaluating the data presented within this review, as well as the fact that most of the studies were completed in healthy animals. It is also important to note that US products approved for use in small ruminants should be used according to the FDA-approved label directions. The FDA-approved label withdrawal time should take precedent above any of the data summarized in this paper.

When considering antibiotic drug classes, it is important to remember that the World Health Organization (WHO) classifies antibiotics into categories based on their place in therapy for some infections in human medicine. These categories include critically important, highly important and important [8]. Some critically important antibiotics are then sub-divided by priority if they are considered sole or limited therapy for some infections in human medicine [8]. Some cephalosporins (third, fourth and fifth generations), quinolones, macrolides are classified as highest priority critically important antibiotics for human health. Aminoglycosides, some cephalosporins (first and second generations) are classified as high priority critically important antibiotics. Amphenicols, some penicillins (antipseudomonal, aminopenicillins with and without beta-lactamase inhibitors, amidinopenicillins, anti-staphylococcal, narrow spectrum), sulfonamides and tetracyclines are classified as highly important antibiotics for human health by the WHO.

### 3.1. Aminoglycosides

Aminoglycosides (amikacin, apramycin, dihydrostreptomycin, gentamicin, tobramycin, neomycin, streptomycin) are concentration dependent, bactericidal antibiotic agents produced from Streptomyces spp. and Micromonospora spp. Aminoglycosides act by irreversibly binding to the 30s subunit of the bacterial ribosome thereby inhibiting protein synthesis. Their spectrum of activity includes mostly Gram-negative bacteria, with some mycobacteria and staphylococci coverage. Transmission of Enterococcus spp., Enterobacteriaceae (including *E. coli*), and Mycobacterium spp. can occur from non-human sources and potentially result in human infection. Therefore, the appropriate use of aminoglycosides in food animal species is essential to maintain human safety. 

Aminoglycosides are generally not well absorbed from the gastrointestinal tract [9], unless there is damage to the intestinal mucosa. When administered parenterally, aminoglycosides are rapidly and completely absorbed. Elimination of aminoglycosides is primarily renally, which may result in persistent residues in the kidneys. In most published studies in sheep and goats, residues in renal tissue exceeded the duration of the study [10,11,12,13,14,15,16,17]. In humans, aminoglycosides are poorly excreted into breastmilk [18]. This may also be the case for sheep and goats as a few studies have shown short duration of residue detection in milk following IV and IM administration [19,20,21,22,23,24,25,26].

In the United States, the only aminoglycoside FDA-approved for use in small ruminants is neomycin sulfate. However, the EMA has approved streptomycin/dihydrostreptomycin and kanamycin for sheep, while also extending MRLs from other species for gentamicin and neomycin. Table 1 shows the published literature that provides data for edible tissue or milk residues of aminoglycosides following treatment of sheep and goats.

### 3.2. Amphenicols

Amphenicols (chloramphenicol, florfenicol, thiamphenicol) are broad-spectrum antibiotics. These antibiotics are typically bacteriostatic agents that act by inhibiting microbial protein synthesis by binding to the 50s bacterial ribosomal subunit. Amphenicols are broad-spectrum against many aerobic and anaerobic Gram-positive and Gram-negative bacteria.

Little pharmacokinetic data is available following the use of amphenicols in sheep or goats. The limited data available in goats shows that florfenicol and thiamphenicol residues do enter the milk after intramuscular and intravenous administration, however tissue data was not available [35,36]. In one study, thiamphenicol concentrations were higher in the mammary gland that was frequently stripped compared to the gland that was not [35].

In the United States, there are no amphenicol products FDA-approved for use in sheep or goats. Chloramphenicol is prohibited from use in food producing animals in several countries including the United States, European Union, and Canada [6,37,38] due to the risk of blood dyscrasias, such as aplastic anemia and bone marrow suppression, in humans. Table 2 summarizes the published literature evaluating edible tissue or milk residues of amphenicols following treatment of sheep and goats.

### 3.3. Penicillin and Penicillin-Derivatives

Penicillins (penicillin G procaine, penicillin G benzathine) and penicillin-derivatives (amoxicillin, ampicillin, cloxacillin, dicloxacillin, nafcillin) are bactericidal antibiotics that act by inhibiting cell wall synthesis. These antibiotics display a broad spectrum of activity against many Gram-positive and Gram-negative bacteria, including anaerobic bacteria.

Amoxicillin and ampicillin show limited milk penetration or accumulation, even when the blood-milk barrier is altered in cases of mastitis [47,48]. However, beta-lactam products labeled for intramammary administration in cattle can result in very high antibiotic concentrations within the small ruminant udder due to the differences in both body and udder size [49,50]. Consequently, intramammary administration of cattle-labeled products to small ruminants can lead to persistent residues present in the milk and require extended withdrawal intervals beyond the labeled withdrawal times for cattle [49,51,52,53,54].In the United States, penicillin G procaine is FDA-approved for use in sheep via intramuscular administration. In the EU, MRLs have been extended from bovine species to all ruminants for nafcillin.

Due to the potential for allergic reactions to penicillin and penicillin-derivatives in humans, caution must be exhibited to ensure food-products from small ruminants do not contain traces of penicillins [55,56]. Table 3 summarizes the published literature evaluating edible tissue or milk residues of beta-lactams or penicillins following treatment of sheep and goats.

### 3.4. Cephalosporins

Cephalosporins (first-generation: cephapirin, cefacetrile, cephalothin, cephradine, cephalexin; second-generation: cefonicid; third-generation: ceftazidime, ceftiofur, ceftriaxone; fourth-generation: cefquinome, cefepime) are beta-lactam antibiotics divided into five ‘generations’ based on the spectrum of activity (first-generation cephalosporins are active against Gram-positive bacteria but not Gram-negative bacteria, while each consecutive generation has increased activity against Gram-negative bacteria with decreased Gram-positive activity). In the United States, cephalosporins are permitted to be used in an extra-label manner in minor species, such as sheep and goats, unlike major food producing species (cattle, swine, chickens & turkeys).

In general, cephalosporins have low penetration into milk [62,63,64,65,66] with variable pharmacokinetic parameters and slower milk depletion in mastitic animals [67,68]. Cephalexin exhibited a nearly double terminal serum elimination half-life in ewes compared to cattle, in addition to increased concentrations of cephalexin residues [69]. Cephapirin exhibited a longer presence of residues in goat samples compared to cattle when used for mastitis treatment [70].

Ceftiofur sodium (Naxcel^®^) is currently the only FDA-approved cephalosporin for use in sheep and goats with a 0 day meat and milk withdrawal time. Pharmacokinetic parameters of both intravenous and intramuscular ceftiofur sodium are found to be similar between sheep and goats when administered at the same dose [71]. Table 4 summarizes the published literature evaluating edible tissue or milk residues of cephalosporins following treatment of sheep and goats.

### 3.5. Fluoroquinolones/Quinolones

Fluoroquinolones (ciprofloxacin, danofloxacin, difloxacin, enrofloxacin, levofloxacin, marbofloxacin, moxifloxacin, norfloxacin, orbifloxacin, pefloxacin, sarafloxacin) are broad-spectrum antibiotics that exhibit concentration-dependent bactericidal activity via inhibition of DNA gyrase in bacterial cells. As a drug class, fluoroquinolones exhibit a high lipid solubility, low protein binding, high bioavailability (especially after parenteral administration) and large volumes of distribution in most species, including small ruminants [81,82,83,84,85,86,87,88,89,90,91,92,93,94,95]. Due to the importance of fluoroquinolones to human health, fluoroquinolones are prohibited from extra-label drug use in food-producing species in the United States.

Studies suggest that the pharmacokinetics of fluoroquinolones change during lactation due to the increased elimination of the drug from serum [88,96]. Additionally, multiple fluoroquinolones extensively penetrate into milk, with some drugs in the class exhibiting up to a 10× higher concentration in milk compared to plasma or serum [88,96,97,98]. This variation can be useful in mastitis cases since these drugs can accumulate in the milk at concentrations above the MIC for a sustained period of time [96,97,99].

In the United States, there are no fluoroquinolones FDA-approved for use in small ruminants, and due to the stipulations outlined by AMDUCA in the CFR, fluoroquinolones are prohibited from extra-label use in food-producing species [6]. In the European Union, flumequine is the only approved fluoroquinolone for use in sheep, while MRLs have been extended from bovine species to all food-producing species for enrofloxacin. Table 5 summarizes the published literature evaluating edible tissue or milk residues of quinolones following treatment of sheep and goats.

### 3.6. Macrolides

Marcolides (erythromycin, gamithromycin, spiramycin, tilmicosin, tulathromycin and tylosin) are a group of bacteriostatic compounds that bind to the 50S bacterial ribosomal subunit inhibiting bacterial protein synthesis and cell growth [110]. These antibiotics are effective against Mycoplasma spp. and Gram-positive organisms, and less effective against Gram-negative organisms.

Penetration into tissues, milk and blood are shown to be relatively quick with high systemic availability [111]. Macrolides show good penetration and distribution into the udder. In particular, tilmicosin and tulathromycin have been shown to have persistent drug residues in the milk [112,113,114,115,116,117], thus they are not recommended for use in lactating animals. Erythromycin, spiramycin and tylosin also exhibit good udder penetration, but result in shorter withdrawal intervals [29,60,111,118,119,120,121,122]. Some small ruminant macrolide pharmacokinetic parameters (absorption, volume of distribution and elimination) were found to be similar to those reported in cattle [111,112,116,123,124].

In the United States, the only FDA-approved macrolide for use in sheep is tilmicosin; however, this approval specifically excludes lactating sheep. Therefore, no tolerance has been established for milk. In the European Union, multiple macrolides are approved for use in small ruminants: gamithromycin and tilmicosin in sheep, and tulathromycin in both sheep and goats. Additionally, MRLs have been extended from other species for erythromycin, tilmicosin (in goats) and tylosin. Table 6 summarizes the published literature evaluating edible tissue or milk residues of macrolides following treatment of sheep and goats.

### 3.7. Sulfonamides

Sulfonamides (sulfadiazine, sulfadimethoxine, sulfamethoxazole, sulfachlorpyrazine) are bacteriostatic antibacterial medications that complete with para-aminobenzoic acid disrupting folic acid synthesis. They are active against Gram-positive and Gram-negative bacteria and protozoa.

One study administered sulfonamides in both normal and mastitic ewes. Sulfonamide concentrations were found to be much higher in the mastitic ewe milk, which the authors attributed in part to the increase in milk pH of mastitic milk [131]. Another study found that some sulfonamides are found in the milk in higher concentrations than blood, whereas others (sulfathiazole, sulfadimidine, sulfadiazine and sulfacetamide) are found in the milk in lower concentrations than blood [132].

Due to the potential for allergic reactions to sulfonamides, caution must be exhibited to ensure food-products from small ruminants do not contain traces of sulfonamides [133,134]. In the US, extra-label use of sulfonamides is prohibited in dairy cattle 20 months of age and older, due to allergic potential of affected milk and increased violative residues.

In the United States, there are no sulfonamide products FDA-approved for use in small ruminants, whereas there are some sulfonamide active ingredients with established milk MRLs for small ruminants in the EU. Table 7 summarizes the published literature evaluating edible tissue or milk residues of sulfonamides following treatment of sheep and goats.

### 3.8. Tetracyclines

Tetracyclines (chlortetracycline, doxycycline oxytetracycline, tetracycline) are broad-spectrum antibiotics that act by inhibiting the 30S bacterial ribosomal subunit thus inhibiting protein synthesis. They are active against Gram-positive and Gram-negative bacteria, as well as some atypical mycobacteria and protozoa [144,145,146].

In oxytetracycline- and chlortetracycline-treated animals, milk production decreased 15% [147]. Infusion of drug into one half of the udder resulted in diffusion of low concentrations into the untreated udder half [147].

Following intramammary infusion of chlortetracycline, residues were detected for a shorter time in goat milk compared to cow milk; however, parenteral chlortetracycline administration results in similar milk residue depletion between goats and cows [122]. 

In the United States, there are multiple tetracycline approvals for both sheep and goats: chlortetracycline (medicated feed for sheep), oxytetracycline (sheep) and tetracycline (sheep and goats; only topical administration for goats). In the EU, MRLs have been determined for chlortetracycline, oxytetracycline and tetracycline in all food-producing species. Table 8 summarizes the published literature evaluating edible tissue or milk residues of tetracyclines following treatment of sheep and goats.

## 4. Conclusions

The judicious use of medications and drug residue avoidance is an important topic in animal agriculture and for veterinarians treating animals that provide food for humans. Although there are numerous published studies that describe drug residues in sheep and goat meat and milk, they are scattered throughout the primary literature. In this review, these studies are compiled, and data extracted for easy reference to help facilitate a comprehensive overview of the scientific data, with respect to drug residues in edible tissues and milk from sheep and goats for antibiotics used in small ruminant practice. 

## Figures and Tables

**Figure 1 animals-12-02607-f001:**
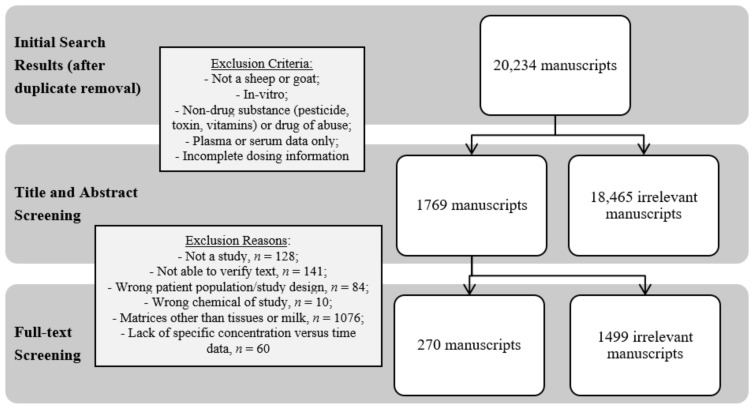
Schematic diagram of the process for three independent evaluators to assess published manuscripts and the numerical outcome of each step. The ultimate goal was to curate manuscripts with tissue and milk residue data from live sheep and goat antibacterial drug studies.

**Table 1 animals-12-02607-t001:** Aminoglycoside residues in milk or edible tissue samples from sheep or goats following treatment.

Analyte	Species; Breed; Age; # of Animals per Time Point	Tolerance/ MRL	Analytical Method	LOD	LOQ	Route of Admin-istration	Dose & Active Ingredient	# of Doses	Matrix	Last Sampling Time Point (Post-Last Treatment) When Residues WERE Detected	Sampling Time Point When NO Residues Were Detected (Post-Last Treatment) *	Health Status	Additional Information	Source/Year
Amikacin	Goat; Baladi; 2–3 years; *n* = 5	US Tol: Not established. EMA MRL: Not established.	Bioassay	NS	0.2 ppm	IV	7.5 mg/kg Amikacin sulfate	1	Milk	4 h (0.22 ppm)	6 h	Healthy	Mid- lactation; Milked 2×	[19] 1999
						IM	7.5 mg/kg Amikacin sulfate	1	Milk	6 h (0.21 ppm)	8 h			
Amikacin	Goat; NS; 1.5–2 years; *n* = 6	US Tol: Not established. EMA MRL: Not established.	Bioassay	0.1 ppm	NS	IM	10 mg/kg Amikacin sulfate	1	Milk	5 h (NS)	6 h	Healthy	Lactating	[20] 2001
Amikacin	Sheep; crossbred; 2–4 years; *n* = 6	US Tol: Not established. EMA MRL: Not established	Bioassay	NS	0.19 ppm	IV	7.5 mg/kg Amikacin sulfate	1	Milk	9.5 h (0.85 ppm ^§^)	>1 day	Healthy	Lactating; Milked 2×/day	[21] 2004
						IM	7.5 mg/kg Amikacin sulfate	1	Milk	9.5 h (0.21 ppm ^§^)	>1 day			
Apramycin	Goat; Saanen; adult; *n* = 10	US Tol: Not established. EMA MRL: Not established.	Bioassay	NS	0.1 ppm	IV	20 mg/kg Apramycin sulfate	1	Milk	10 h (0.12 ppm ^§^)	>10 h	Healthy	Early Lactation	[22] 1995
Apramycin	Sheep; Awassi; adult; *n* = 6	US Tol: Not established. EMA MRL: Not established.	Bioassay	NS	0.1 ppm	IM	10 mg/kg Apramycin sulfate	1	Milk	720 min (0.15 ppm ^§^)	1440 min	Diseased-Mastitis	Mid-lactation	[22] 1995
Apramycin	Sheep; Awassi; adult; *n* = 10	US Tol: Not established. EMA MRL: Not established.	Bioassay	NS	0.1 ppm	IV	20 mg/kg Apramycin sulfate	1	Milk	6 h (0.11 ppm ^§^)	8 h	Healthy	Mid-lactation	[22] 1995
Apramycin	Sheep; NS; Lambs; *n* = 12 study; *n* = 3/time pt	US Tol: Not established. EMA MRL: Not established.	Bioassay	500 ppb	NS	PO	10 mg/kg Apramycin daily	3	Liver	ND@1 day	1 day	Healthy	NS	[10] 1999
									Kidney	21 days (1730 ppb)	35 days			
									Muscle	ND@1 day	1 day			
									Fat	21 days (960 ppb)	28 days			
	NS; Lambs; *n* = 20 study; *n* = 4/time pt		HPLC	Liver: 368 ppb	Liver: 2500 ppb	PO	10 mg/kg Apramycin daily	5	Liver	30 days (700 ppb)	>30 days			
				Kidney: 394 ppb	Kidney: 2500 ppb				Kidney	30 days (1700 ppb)	>30 days			
				Muscle: 124 ppb	Muscle: 500 ppb				Muscle	ND @ 6 days	6 days			
				Fat: 42 ppb	Fat: 500 ppb				Fat	ND @ 6 days	6 days			
Apramycin	Goat; NS; Adult; NS	US Tol: Not established. EMA MRL: Not established.	NS	NS	NS	IM	20 mg/kg Apramycin	1	Milk	10 h (NS)	12 h	NS	NS	[23] 2000
						IV	20 mg/kg Apramycin	1	Milk	12 h (NS)	>12 h			
Dihydro- strepto-mycin	Goat; NS; Adults; *n* = 220	US Tol: Not established. EMA MRL: Not established.	Bioassay	0.13 ppm	0.15 ppm	IMM	300,000 IU Procaine benzyl-penicillin; 100 mg dihydro-strepto-mycin; 100 mg nafcillin	1	Milk	6 days post kidding (≥0.2 ppm)	7 days post kidding	Healthy	Dry off period (mean 61.0 ± 4.3 days SD (range 23–156 days); 1 tube per gland before drying off. Sample collected after kidding	[27] 1995
Dihydro-strepto-mycin	Sheep; Lacaune; adult; *n* = 8	US Tol: Not established. EMA established MRL: 200 ppb (milk).	Bioassay	0.02 ppm	NS	IMM	300,000 IU Procaine benzyl-penicillin; 100 mg dihydro-streptomycin; 100 mg nafcillin	1	Milk	3 days (0.02 ppm ^§^)	4 days	Healthy	Dry off period (mean 112 days (range 85–223 days); 1 tube per gland before drying off. Sample collected after lambing	[28] 1995
Dihydro-strepto-mycin	Sheep; Awassi; adult; *n* = 3	US Tol: Not established. EMA established MRL: 200 ppb (milk).	Bioassay	NS	NS	IV	20 mg/kg Dihydro-streptomycin (radio-labeled) then 10 mg/kg for 4 doses 45 min interval	5	Milk	24 h (0.20 ppm ^§^)	36 h	Healthy	Lactating; Milked 2×/day	[24] 1973
			Radio-activity	NS	NS	IV		5	Milk	8 h (1.83 ppm ^§^)	10 h			
Dihydro-strepto-mycin	Sheep; NS; NS; *n* = 22 study *n* = 4/time pt	US Tol: Not established. EMA established MRL: 500 ppb (liver, muscle, fat); 1000 ppb (kidney).	Bioassay	0.5 ppm	NS	IM	10 mg/kg Dihydro-streptomycin combined w/ 10,000 IU procaine penicillin-G daily	5	Kidney	28 days (0.8 ppm)	>28 days	Healthy	NS	[11] 1995
									Muscle	14 days (0.07 ppm)	21 days			
									Inj. Site	28 days (0.2 ppm)	>28 days			
Dihydro-strepto-mycin	Sheep; Awassi; Adult; *n* = 2	US Tol: Not established. EMA established MRL: 200 ppb (milk).	Bioassay	NS	NS	IM	20 mg/kg Dihydro-streptomycin (radio-labeled)	1	Milk	12 h (0.22 ppm ^§^)	24 h	Healthy	Lactating	[29] 1974
									Milk	48 h (0.11 ppm ^§^)	56 h	Disease-mastitis		
			Radio-activity	NS	NS	IM			Milk	48 h (0.75 ppm ^§^)	>48 h	Healthy		
									Milk	12 h (0.42 ppm ^§^)	24 h	Disease-mastitis		
Dihydro-strepto-mycin	Sheep; NS; NS; *n* = 12 study; *n* = 4/ time pt NS; Adult; *n* = 8	US Tol: Not established. EMA established MRL: 500 ppb (liver, muscle, fat); 1000 ppb (kidney); 200 ppb (milk).	NS	NS	NS	IM	10 mg/kg Dihydro-streptomycin combined with benzyl-penicillin daily	3	Liver	<400 ppb @ 14 days	14 days	Healthy	NS	[25] 2005
									Kidney	<400 ppb @ 14 days	14 days			
									Muscle	<400 ppb @ 14 days	14 days			
									Fat	<400 ppb @ 14 days	14 days			
									Inj. Site	18 days (0.58 ppm)	28 days			
			HPLC	NS	50 ppb	IM	10 mg/kg Dihydro-streptomycin sulfate combined with 10 mg/kg streptomycin daily	3	Milk	48 h (0.06 ppm)	60 h	Healthy	Lactating	
Dihydro-strepto-mycin	Sheep; Suffolk & Suffolk/Cheviot; adult; *n* = 8	US Tol: Not established. EMA established MRL: 200 ppb (milk).	HPLC	0.02 ppm	0.05 ppm	IM	10 mg/kg Dihydro-streptomycin combined with 10 mg/kg streptomycin daily	3	Milk	48 h (0.06 ppm)	60 h	Healthy	Lactating; Milked 2×/day	[26] 2002
Dihydro-strepto-mycin	Sheep; NS; NS; *n* = 12 study; *n* = 4/time pt	US Tol: Not established. EMA established MRL: 500 ppb (liver, muscle, fat); 1000 ppb (kidney).	NS	NS	400 ppb	IM	10 mg/kg Dihydro-streptomycin combined w/ benzyl-penicillin daily	3	Liver	<LOQ @ 14 days	14 days	Healthy	NS	[30] 2000
									Kidney	<LOQ @ 14 days	14 days			
									Muscle	<LOQ @ 14 days	14 days			
									Fat	<LOQ @ 14 days	14 days			
									Inj. Site	18 days (0.584 ppm)	28 days			
Dihydro-strepto-mycin	Sheep; NS; NS; *n* = 12 study; *n* = 4/time pt	US Tol: Not established. EMA established MRL: 500 ppb (liver, muscle, fat); 1000 ppb (kidney).	HPLC	NS	400 ppb	IM	10 mg/kg Dihydro-streptomycin combined w/ procaine penicillin daily	3	Liver	<LOQ @ 14 days	14 days	Healthy	NS	[31] 1998
									Kidney	<LOQ @ 14 days	14 days			
									Muscle	<LOQ @ 14 days	14 days			
									Fat	<LOQ @ 14 days	14 days			
									Inj. Site	18 days (0.584 ppm)	28 days			
Gentamicin	Sheep; mixed breed; adult; *n* = 7	US Tol: Not established. EMA established MRL in all mammalian food producing species: 750 ppb (kidney).	Bioassay	NS	NS	IV	4 mg/kg Gentamicin	1	Kidney *biopsy	28 days (9.9 ppm)	35 days	Healthy	NS	[32] 1985
Gentamicin	Sheep; Suffolk; adult; *n* = 9 study; *n* = 3/time pt	US Tol: Not established. EMA established MRL in all mammalian food producing species: 750 ppb (kidney); 50 ppb (muscle).	Immuno-assay	0.01 ppm	NS	IM	3 mg/kg Gentamicin sulfate at 8 h intervals	2	Kidney	15 days (20.0 ppm ^§^)	>15 days	Healthy	NS	[12] 1985
									Muscle	15 days (0.21 ppm ^§^)	>15 days			
									Heart	15 days (0.64 ppm ^§^)	>15 days			
Gentamicin	Sheep; Suffolk; adult; *n* = 12 study; *n* = 3/time pt	US Tol: Not established. EMA Established MRL in all mammalian food producing species: 200 ppb (liver); 750 ppb (kidney); 50 ppb (muscle, fat).	Immuno-assay	0.01 ppm	NS	IM	2 mg/kg Gentamicin sulfate	1	Liver	12 days (0.31 ppt)	>12 days	Healthy	NS	[13] 1986
									Kidney	12 days (2.74 ppt)	>12 days			
									Muscle	12 days (0.2 ppt)	>12 days			
									Inj. Site	12 days (0.15 ppt)	>12 days			
							6 mg/kg Gentamicin sulfate	1	Liver	12 days (1.5 ppt)	>12 days			
									Kidney	12 days (5.15 ppt)	>12 days			
									Muscle	12 days (0.002 ppt)	>12 days			
									Inj. Site	12 days (0.02 ppt)	>12 days			
							18 mg/kg Gentamicin sulfate	1	Liver	12 days (4.0 ppt)	>12 days			
									Kidney	12 days (9.23 ppt)	>12 days			
									Muscle	12 days (0.14 ppt)	>12 days			
									Inj. Site	12 days (0.53 ppt)	>12 days			
							2 mg/kg Gentamicin sulfate at 8 h intervals	9	Liver	12 days (4.02 ppt)	>12 days			
									Kidney	12 days (9.74 ppt)	>12 days			
									Muscle	12 days (0.04 ppt)	>12 days			
									Inj. Site	12 days (2.49 ppt)	>12 days			
							6 mg/kg Gentamicin sulfate daily	3	Liver	12 days (3.12 ppt)	>12 days			
									Kidney	12 days (10.0 ppt)	>12 days			
									Muscle	12 days (0.14 ppt)	>12 days			
									Inj. Site	12 days (5.03 ppt)	>12 days			
Gentamicin	Sheep; western range; adult; *n* = 4	US Tol: Not established. EMA established MRL in all mammalian food producing species: 750 ppb (kidney).	Immuno-assay	0.04 ppm	NS	IM	3 mg/kg Gentamicin sulfate at 12 h intervals	20	Kidney (biopsy)	77 days (9.71 ppm)	>77 days	NS	NS	[14] 1988
Kanamycin	Sheep; Bergamo; adult; *n* = 12 study; *n* = 3/time pt	US Tol: Not established. EMA established MRL: 600 ppb (liver); 2500 ppb (kidney); 100 ppb (muscle).	Bioassay	NS	NS	IM	20 mg/kg Kanamycin	1	Liver	3 days (2.2 ppm)	6 days	NS	NS	[33] 1991
									Kidney	10 days (8.31 ppm)	14 days			
									Muscle	ND @ 3 days	3 days			
Neomycin	Goat; NS; NS; *n* = 18 study; *n* = 4/ time pt	US Tol: 3600 ppb (liver); 7200 ppb (kidney);1200 ppb (muscle); 7200 ppb (fat). EMA MRL extrapolated to all food producing species: 5500 ppb (liver); 9000 ppb (kidney); 500 ppb (muscle, fat).	Bioassay	NS	0.5 ppm	POMW	22 mg/kg Neomycin sulfate daily	14	Liver	ND @ 12 h	12 h	Healthy	NS	[15] 1996
									Kidney	96 h (0.6 ppm)	>96 h			
									Muscle	ND @ 12 h	12 h			
									Fat	ND @ 12 h	12 h			
Neomycin	Goat; NS; NS; *n* = 20 study; *n* = 5/time pt	US Tol: 3600 ppb (liver); 7200 ppb (kidney); 1200 ppb (muscle); 7200 ppb (fat). EMA MRL extrapolated to all food producing species: 5500 ppb (liver); 9000 ppb (kidney); 500 ppb (muscle, fat).	Bioassay	NS	500 ppb	POMW	20 mg/kg Neomycin sulfate daily	14	Liver	ND @ 12 h	12 h	Healthy	NS	[16] 2000
									Kidney	96 h (700 ppb)	>96 h			
									Muscle	ND @ 12 h	12 h			
									Fat	ND @ 12 h	12 h			
Neomycin	Goat; NS; NS; *n* = 20 study, *n* = 4/time pt	US Tol: 3600 ppb (liver); 7200 ppb (kidney); 1200 ppb (muscle); 7200 ppb (fat). EMA MRL extrapolated to all food producing species: 5500 ppb (liver); 9000 ppb (kidney); 500 ppb (muscle, fat).	Bioassay	NS	0.5 ppm	PO	22 mg/kg Neomycin sulfate daily	14	Liver	ND @ 12 h	12 h	Healthy	NS	[17] 1995
									Kidney	96 h (0.7 ppm)	>96 h			
									Muscle	ND @ 12 h	12 h			
									Fat	ND @ 12 h	12 h			
Neomycin	Goat; NS; NS; *n* = 20 study, *n* = 4/time pt	US Tol: 3600 ppb (liver); 7200 ppb (kidney); 1200 ppb (muscle); 7200 ppb (fat). EMA MRL extrapolated to all food producing species: 5500 ppb (liver); 9000 ppb (kidney); 500 ppb (muscle, fat).	NS	NS	0.5 ppm	POMW	22 mg/kg Neomycin sulfate daily	14	Liver	ND @ 12 h	12 h	Healthy	NS	[34] 1996
									Kidney	96 h (0.57 ppm)	>96 h			
									Muscle	ND @ 12 h	12 h			
									Fat	ND @ 12 h	12 h			
Neomycin	Sheep; NS; NS; *n* = 18 study; *n* = 4/time pt	US Tol: 3600 ppb (liver); 7200 ppb (kidney); 1200 ppb (muscle); 7200 ppb (fat). EMA MRL extrapolated to all food producing species: 5500 ppb (liver); 9000 ppb (lidney); 500 ppb (muscle, fat).	Bioassay	NS	0.5 ppm	POMW	22 mg/kg Neomycin sulfate daily	14	Liver	ND @ 1 day	1 day	Healthy	NS	[15] 1996
									Kidney	1 day (female) (1.28 ppm)	3 days (female)			
									Kidney	3 days (male) (0.45 ppm)	7 days (male)			
									Muscle	ND @ 1 day	1 day			
									Fat	ND @ 1 day	1 day			
Neomycin	Sheep; NS; NS; *n* = 20 study; *n* = 5/time pt	US Tol: 3600 ppb (liver); 7200 ppb (kidney); 1200 ppb (muscle); 7200 ppb (fat). EMA MRL extrapolated to all food producing species: 5500 ppb (liver); 9000 ppb (kidney); 500 ppb (muscle, fat).	Bioassay	NS	500 ppb	POMW	20 mg/kg Neomycin sulfate daily	14	Liver	ND @ 1 day	1 day	Healthy	NS	[16] 2000
									Kidney	3 days (522 ppb)	7 days			
									Muscle	ND @ 1 day	1 day			
									Fat	ND @ 1 day	1 day			
Neomycin	Sheep; NS; NS; *n* = 20 study, *n* = 4/time pt	US Tol: 3600 ppb (liver); 7200 ppb (kidney); 1200 ppb (muscle); 7200 ppb (fat). EMA MRL extrapolated to all food producing species: 5500 ppb (liver); 9000 ppb (kidney); 500 ppb (muscle, fat).	Bioassay	NS	0.5 ppm	PO	22 mg/kg Neomycin sulfate daily	14	Liver	ND @ 1 day	1 day	Healthy	NS	[17] 1995
									Kidney	3 days (522 ppb)	7 days			
									Muscle	ND @ 1 day	1 day			
									Fat	ND @ 1 day	1 day			
Neomycin	Sheep; NS; NS; *n* = 20 study, *n* = 4/time pt	US Tol: 3600 ppb (liver); 7200 ppb (kidney); 1200 ppb (muscle); 7200 ppb (fat). EMA MRL extrapolated to all food producing species: 5500 ppb (liver); 9000 ppb (kidney); 500 ppb (muscle, fat).	NS	NS	0.5 ppm	POMW	22 mg/kg Neomycin sulfate daily	14	Liver	ND @ 1 day	1 day	Healthy	NS	[34] 1996
									Kidney	1 day (female) (1.28 ppm)	3 days (female)			
									Kidney	3 days (male) (0.45 ppm)	7 days (male)			
									Muscle	ND @ 1 day	1 day			
									Fat	ND @ 1 day	1 day			
Strepto-mycin	Sheep; NS; NS; *n* = 4	US Tol: Not established. EMA established MRL: 200 ppb (milk).	HPLC	NS	50 ppb	IM	10 mg/kg Streptomycin combined w/ dihydro-streptomycin daily	3	Milk	48 h (0.07 ppm)	60 h	Healthy	Lactating, Milked 2×/day	[25] 2005
Strepto-mycin	Sheep; Suffolk & Suffolk/Cheviot; adult; *n* = 8	US Tol: Not established. EMA established MRL: 200 ppb (milk).	HPLC	0.02 ppm	0.05 ppb (^†^ 0.05 ppm)	IM	10 mg/kg Streptomycin combined w/ dihydro-streptomycin daily	3	Milk	48 h (0.07 ppm)	60 h	Healthy	Milked 2×/day	[26] 2002
Strepto-mycin	Sheep; NS; NS; NS	US Tol: Not established. EMA established MRL: 500 ppb (liver, muscle, fat); 1000 ppb (kidney).	HPLC	NS	200 ppb	IM	10 mg/kg Streptomycin daily	3	Liver	2 days (655 ppb)	>2 days	Healthy	NS	[30] 2000
									Kidney	2 days (914 ppb)	>2 days			
									Muscle	ND @ 2 days	2 days			
									Fat	ND @ 2 days	2 days			
									Inj. Site	2 days (1373 ppb)	>2 days			

^†^ Manuscript states limit of quantitation as 0.05 ppb; however, the limit of detection is in parts per million, therefore it is likely an error and should be interpreted as 0.05 parts per million. # = number. * Projected time for which residues could still be detected based on study protocol for sample collection time points and sample concentration results. Authors caution readers to critically evaluate these publications to estimate when full residue depletion might occur. Abbreviations: 2×/day: twice daily. LOD: Limit of detection. LOQ: Limit of quantification. EMA: European Medicines Agency. MRL: Maximum residue limit. ND: Not detected. NS: Not specified. Routes of Administration: IMM = intramammary, IM = intramuscular, IV = intravenous, PO = per os, POMF = per os as medicated feed, POMW = per os as medicated water, SC = subcutaneous. ^§^ Data points manually extracted use scanning software (Webplot digitizer or UnScanIt 7.0). Units: s = seconds, min = minutes, h = hours, ppt = parts per trillion, ppb = parts per billion, ppm = parts per million.

**Table 2 animals-12-02607-t002:** Amphenicol residues in milk or edible tissue samples from sheep or goats following treatment.

Analyte	Species; Breed; Age; # of Animals per Time Point	Tolerance/ MRL	Analytical Method	LOD	LOQ	Route of Admin-istration	Dose & Active Ingredient	# of Doses	Matrix	Last Sampling Time Point (Post-Last Treatment) When Residues WERE Detected	Sampling Time Point When NO Residues Were Detected (Post-Last Treatment) *	Health Status	Additional Information	Source/ Year
Chloram-phenicol	Sheep; Awassi; adult; *n* = 2	US Tol: Not established. EMA MRL: Not established.	Chemically	NS	NS	IM	50 mg/kg Chloramphenicol	1	Milk	26 h (1.68 ppm ^§^)	>26 h	Healthy	Lactating	[39] 1973
									Milk	26 h (1.82 ppm ^§^)	>26 h	Diseased-mastitis		
			Bioassay	NS	NS	IM	50 mg/kg Chloramphenicol	1	Milk	26 h (1.02 ppm ^§^)	>26 h	Healthy		
									Milk	26 h (1.54 ppm ^§^)	>26 h	Diseased-mastitis		
			Radio-activity	NS	NS	IM	50 mg/kg Chloramphenicol (radiolabeled)	1	Milk	13 h (NS)	>13 h	Healthy		
									Milk	13 h (NS)	>13 h	Diseased-mastitis		
Chloramphenicol	Sheep; Awassi; adult; *n* = 1	US Tol: Not established. EMA MRL: Not established.	Bioassay	NS	NS	IV	50 mg/kg Chloramphenicol sodium succinate then 12.5 mg/kg for 2 doses at 90 min interval	3	Milk	24 h (0.65 ppm ^§^)	36 h	Healthy	Lactating; Milked 2x/day	[24] 1973
			Radio-activity	NS	NS	IV	50 mg/kg Chloramphenicol (radiolabeled) then 12.5 mg/kg for 2 doses at 90 min interval	3	Milk	48 h (0.81 ppm ^§^)	60 h			
Chloram-phenicol	Sheep; Awassi; Adult; *n* = 2	US Tol: Not established. EMA MRL: Not established.	Bioassay	NS	NS	IM	50 mg/kg Chloramphenicol	1	Milk	56 h (0.85 ppm ^§^)	>56 h	Healthy	Lactating; Milked 2×/day	[29] 1974
									Milk	56 h (1.28 ppm ^§^)	>56 h	Diseased-mastitis		
			Radio-activity	NS	NS	IM	50 mg/kg Chloramphenicol (radiolabeled)	1	Milk	56 h (0.2 ppm ^§^)	>56 h	Healthy		
									Milk	56 h (0.18 ppm ^§^)	>56 h	Diseased-mastitis		
Chloram-phenicol	Sheep; Rouge de L’Ouest; adult; *n* = 11 study; *n* = 3 & 2/time pt	US Tol: Not established. EMA MRL: Not established.	HPLC	2 ppb	NS	IM	30 mg/kg Chloramphenicol	1	Liver	24 h (0.35 ppb ^§^)	336 h	NS	NS	[40] 1990
									Kidney	336 h (0.76 ppb ^§^)	>336 h			
									Muscle	336 h (2.13 ppb ^§^)	>336 h			
									Inj. Site	336 h (4.18 ppb ^§^)	>336 h			
Chloram-phenicol	Sheep; Awassi; adult; *n* = 2	US Tol: Not established. EMA MRL: Not established.	Bioassay	NS	NS	IV	50 mg/kg Chloramphenicol sodium succinate	1	Milk	NS	NS	Healthy	Lactating; Milked 2×/day	[41] 1975
Florfenicol amine	Sheep; Polypay; NS; *n* = 25 study; *n* = 5/time pt	US Tol: Not established. EMA MRL by extension from bovine to ovine: 3000 ppb (liver); 300 ppb (kidney); 200 ppb (muscle).	HPLC	NS	NS	SC	40 mg/kg Florfenicol daily	3	Liver	40 days (1.99 ppm)	>40 days	NS	NS	[42] 2006
									Kidney	40 days (0.17 ppm)	>40 days			
									Muscle	40 days (0.08 ppm)	>40 days			
									Fat	40 days (0.01 ppm)	>40 days			
									Inj. Site	40 days (0.15 ppm)	>40 days			
Florfenicol amine	Sheep; mixed breed; 6–7 months; *n* = 26 study; *n* = 5/time pt	US Tol: Not established. EMA MRL by extension from bovine to ovine: 3000 ppb (liver); 300 ppb (kidney); 200 ppb (muscle).	HPLC	NS	Liver: 0.32 ppm	SC	40 mg/kg Florfenicol daily	3	Liver	40 days (NS)	>40 days	Healthy	NS	[43] 2008
					Kidney: 0.1 ppm				Kidney	40 days (NS)	>40 days			
					Muscle: 0.05 ppm				Muscle	40 days (NS)	>40 days			
					Fat: 0.04 ppm				Fat	40 days (NS)	>40 days			
					Inj. Site: 0.05 ppm				Inj. Site	40 days (NS)	>40 days			
Thiam-phenicol	Sheep; crossbred; adult; *n* = 16 study; *n* = 4/time pt.	US Tol: Not established. EMA MRL by extension from bovine to ovine: 50 ppb (liver, kidney, muscle, fat, milk).	HPLC	5 ppb	21 ppb	IM	30 mg/kg Thiamphenicol daily	5	Liver	ND @ 4 days	4 days	Healthy	NS	[44] 2000
									Kidney	4 days (40.2 ppb)	8 days			
									Muscle	<LOD @ 4 days	4 days			
									Fat	4 days (342.5 ppb)	8 days			
Chloram-phenicol	Goat; Desi; 9–12 months; *n* = NS	US Tol: Not established. EMA MRL: Not established.	Colorimetric	NS	NS	IM	10 mg/kg Chloramphenicol	1	Milk	24 h (2.16 ppm)	2 days	Healthy	Lactating	[45] 1983
						IM	30 mg/kg Chloramphenicol	1	Milk	96 h (3.33 ppm)	>4 days	Healthy		
Chloram-phenicol	Goat; NS; Adult; *n* = 2	US Tol: Not established. EMA MRL: Not established.	HPLC	5 ppb	NS	IM	600 mg Chloram-phenicol	1	Milk	8 h (0.077 ppm)	1 day	Healthy	Lactating	[46] 1980
						IMM	600 mg Chloram-phenicol	1	Milk	24 h (0.026 ppm)	32 h	Healthy		
Thiam-phenicol	Goat; Saanen & crossbred; adult; *n* = 6	US Tol: Not established. EMA MRL by extension from bovine to ovine: 50 ppb (liver, kidney, muscle, fat, milk).	HPLC	NS	NS	IV	50 mg/kg Thiamphenicol	1	Milk	12 h (4.92 ppm ^§^)	>12 h	Healthy	Late lactation	[35] 1991
						IM	50 mg/kg Thiamphenicol	1	Milk	12 h (4.90 ppm ^§^)	>12 h	Healthy		
Florfenicol	Goat; Saanen & crossbred; adult; *n* = 10	US Tol: Not established. EMA MRL: Not established.	HPLC	NS	NS	IV	25 mg/kg Florfenicol	1	Milk	8 h (0.21 ppm ^§^)	>8 h	Healthy	Mid-lactation	[36] 1991
						IM	25 mg/kg Florfenicol	1	Milk	8 h (0.11 ppm ^§^)	>8 h	Healthy		

^§^ Data points manually extracted use scanning software (Webplot digitizer or UnScanIt 7.0). # = number. * Projected time for which residues could still be detected based on study protocol for sample collection time points and sample concentration results. Authors caution readers to critically evaluate these publications to estimate when full residue depletion might occur. Abbreviations: 2×/day: twice daily. LOD: Limit of detection. LOQ: Limit of quantification. EMA: European Medicines Agency. MRL: Maximum residue limit. ND: Not detected. NS: Not specified. Routes of Administration: IMM = intramammary, IM = intramuscular, IV = intravenous, PO = per os, POMF = per os as medicated feed, SC = subcutaneous. Units: s = seconds, min = minutes, h = hours, ppb = parts per billion, ppm = parts per million.

**Table 3 animals-12-02607-t003:** Penicillin and penicillin-derivative residues in milk or edible tissue samples from sheep or goats following treatment.

Analyte	Species; Breed; Age; # of Animals	Tolerance/MRL	Analytical Method	LOD	LOQ	Route of Admin-istration	Dose & Active Ingredient	# of Doses	Matrix	Last Sampling Time Point (Post-Last Treatment) When Residues WERE Detected	Sampling Time Point When NO Residues Were Detected (Post-Last Treatment) *	Health Status	Additional Information	Source/ Year
Amoxicillin	Sheep; Texel; adult; *n* = 12	US Tol: Not established. EMA MRL: Not established.	Bioassay	NS	NS	IM	10 mg/kg Amoxicillin sodium	1	Milk	500 min (0.03 ppm ^§^)	>500 min	Healthy & Diseased- mastitis	Lactating	[47] 1979
Amoxicillin	Goats; Saanen; adult; *n* = 6	US Tol: Not established. EMA MRL: Not established.	Bioassay	0.001 ppm	NS	IMM	200 mg Amoxicillin trihydrate; 50 mg potassium clavulanate; 10 mg prednisolone combo product at 8 h intervals	3	Milk	5 days (0.07 ppm ^§^)	>5 days	Healthy	Lactating; Milked 2×/day; 1 syringe/ gland	[49] 1989
Amoxicillin	Sheep; Texel; adult; *n* = 12	US Tol: Not established. EMA MRL: Not established.	Bioassay	NS	NS	IM	10 mg/kg Amoxicillin trihydrate	1	Milk	500 min (0.06 ppm ^§^)	>500 min	Healthy & Diseased- mastitis	Lactating	[47] 1979
Amoxicillin	Sheep; Friesland; adult; *n* = 6	US Tol: Not established. EMA MRL: Not established.	Bioassay	0.001 ppm	NS	IMM	200 mg Amoxicillin trihydrate; 50 mg potassium clavulanate; 10 mg prednisolone combo product at 8 h intervals	3	Milk	7 days (0.003 ppm ^§^)	>7 days	Healthy	Lactating; Milked 2×/day; 1 syringe/ gland	[51] 1989
Amoxicillin	Sheep; Comisana; adult; *n* = 10	US Tol: Not established. EMA MRL: Not established.	HPLC	1.5 ppb	2.5 ppb	IM	12.5 mg/kg Amoxicillin trihydrate (long acting)	1	Milk	132 h (1.5 ppb)	6 days	Healthy	Lactating; Milked 2×/day	[57] 2002
Amoxicillin	Sheep; domestic dairy breed; adult; *n* = 10	US Tol: Not established. EMA MRL: Not established.	Bioassay	3 ppb	4 ppb	IMM w/ IM	200 mg Amoxicillin trihydrate, 50 mg potassium clavulanate, 10 mg prednisolone combination product (IMM) at 12 h intervals co-administered with 140 mg/35 mg per mL amoxicillin trihydrate/ clavulanic acid (IM) at 24 h intervals	5 (IMM); 2(IM)	Milk	192 h (4.5 ppb)	>192 h	Diseased- mastitis	Lactating; 1 syringe/gland	[52] 2009
Amoxicillin	Sheep; crossbred; NS; *n* = 36 study; *n* = 4/time pt Dairy type; adult; *n* = 20	US Tol: Not established. EMA MRL: Not established.	LC-MS	5.8 ppb	25.6 ppb	IM	7 mg/kg Amoxicilllin ^†^ daily	5	Liver	NS	48 h	Healthy	NS	[48] 2012
									Kidney	NS	48 h			
									Muscle	NS	48 h			
									Fat	NS	48 h			
									Inj. Site	64 days (25.6 ppb)	>64 days			
			NS	NS	NS	IM	7 mg/kg Amoxicilllin ^†^ daily	5	Milk	120 h (2.09 ppb)	>120 h	Healthy	Lactating; Milked 2×/day	
Ampicillin	Sheep; Texel; adult; *n* = 12	US Tol: Not established. EMA MRL: Not established.	Bioassay	NS	NS	IM	10 mg/kg Ampicillin sodium	1	Milk	8 h (0.03 ^§^)	>8 h	Healthy	Lactating	[47] 1979
										10 h (0.03 ^§^)	>10 h	Diseased- mastitis		
Ampicillin	Goats; Saanen; adult; *n* = 24 study	US Tol: Not established. EMA MRL: Not established.	HLPC	1.5 ppb	2.2 ppb	IM	15 mg/kg Amoxicilllin ^†^ (long acting) at 72 h interval	2	Milk	168 h (6.0 ppb)	180 h	Healthy	Mid-lactation; Milked 2×/day	[54] 2010
Ampicillin	Sheep; Texel; adult; *n* = 12	US Tol: Not established EMA MRL: Not established.	Bioassay	NS	NS	IM	10 mg/kg Ampicillin trihydrate	1	Milk	12 h (0.04 ppm ^§^)	>12 h	Healthy	Lactating	[47] 1979
										12 h (0.1 ppm ^§^)	>12 h	Diseased- mastitis		
Ampicillin	Sheep; NS; adult; *n* = 4	US Tol: Not established. EMA MRL: Not established.	NS	NS	NS	IMM	250,000 IU Ampicillin trihydrate	1	Milk	72 h (0.11 ppm)	96 h	NS	Lactating; Half syringe per gland	[58] 1977
Cloxacillin	Goats; Saanen; adult; *n* = 8	US Tol: Not established. EMA MRL: Not established.	Bioassay	0.02 ppm	NS	IMM	200 mg Cloxacillin at 48 h intervals	3	Milk	13 h (0.15 ppm^§^)	>13 h	Healthy	Late lactation; Milked 2×/day. Only one half/gland treated.	[53] 1984
Diclox-acillin	Sheep; Sarda; 2–3.5 years; *n* = 4	US Tol: Not established. EMA MRL: Not established.	HPLC	NS	0.02 ppm	IMM	100 mg/half Dicloxacillin at 12 h intervals.	3	Milk	60 h (0.029 ppm)	72 h	Healthy	Lactating, High production; Milked 2x/day	[50] 2000
										72 h (0.026 ppm)	84 h	Healthy	Lactating, Low production; Milked 2×/day	
Nafcillin	Goats; NS; Adults; *n* = 220	US Tol: Not established. EU MRL by extension from bovine to all ruminants: 30 ppb (milk).	Bioassay	0.012 ppm	0.015 ppm	IMM	300,000 IU Procaine benzylpenicillin; 100 mg dihydro-streptomycin; 100 mg nafcillin	1	Milk	NS	3 days	Healthy	Dry off period (mean 61.0 ± 14.3 days SD (range 23–156 days); 1 tube per gland before drying off. Sample collected after kidding	[27] 1995
Nafcillin	Sheep; Lacaune; adult; *n* = 8	US Tol: Not established. EMA MRL by extension from bovine to all ruminants: 30 ppb (milk).	Bioassay	0.02 ppm	NS	IMM	300,000 IU Procaine benzylpenicillin; 100 mg dihydrostreptomycin; 100 mg nafcillin	1	Milk	ND	2 days	Healthy	Dry off period (mean 112 days (range 85–223 days); 1 tube per gland before drying off. Sample collected after lambing	[28] 1995
Pen-ethamate	Goats; NS; Adult; *n* = 2	US Tol: Not established. EMA MRL: Not established.	Bioassay	NS	NS	IM	200,000 IU Penethamate (oil) 200,000 IU Penethamate (aqueous)	1	Milk	1 day (0.004 U/mL)	>1 day	NS	Lactating	[59] 1966
									Milk	12 h (0.075 U/mL)	1 day			
						IM	500,000 IU Penethamate (oil) 500,000 IU Penethamate (aqueous)	1	Milk	1 day (0.04 U/mL)	>1 day			
									Milk	1 day (0.2 U/mL)	>1 day			
Penicillin	Sheep; Awassi; Adult; *n* = 2	US Tol: Not established. EMA MRL: Not established.	Bioassay	NS	NS	IM	20 mg/kg Penicillin ^†^	1	Milk	12 h (0.02 ppm ^§^)	1 day	Healthy	Lactating	[29] 1974
			Radioactivity	NS	NS				Milk	56 h (0.03 ppm ^§^)	>56 h	Diseased- mastitis		
			Bioassay	NS	NS	IM	20 mg/kg Benzylpenicillin-14C	1	Milk	48 h (0.01 ppm ^§^)	56 h	Healthy		
			Radioactivity	NS	NS				Milk	12 h (0.02 ppm ^§^)	1 day	Diseased- mastitis		
Penicillin	Goats; NS; Adult; *n* = 2	US Tol: Not established. EMA MRL: Not established.	Bioassay	NS	NS	IM	200,000 IU Procaine penicillin (oil)	1 1	Milk	1 day (0.008 U/mL)	>1 day	NS	Lactating	[59] 1966
							200,000 IU Procaine penicillin (aqueous)		Milk	12 h (0.012 U/mL)	1 day			
						IM	500,000 IU Procaine penicillin (oil)		Milk	1 day (0.07 U/mL)	>1 day			
							500,000 IU Procaine penicillin (aqueous)		Milk	1 day (0.02 U/mL)	>1 day			
Penicillin	Goats; NS; Adults; *n* = 217	US Tol: Not established. EMA MRL: Not established.	Bioassay	0.002 IU/mL	0.004 IU/mL	IMM	300,000 IU Procaine benzylpenicillin; 100 mg dihydro-streptomycin; 100 mg nafcillin combo product	1	Milk	NS	7 days	Healthy	Dry off period (mean 61.0 ± 14.3 days SD (range 23–156 days). 1 tube per gland before drying off. Sample collected after kidding	[27] 1995
Penicillin	Goats; dairy type; 2–7 years; *n* = 10	US Tol: Not established. EMA MRL: Not established.	Bioassay	NS	NS	IMM	100,000 IU Penicillin G procaine at 12 h intervals	3	Milk	60 h (0.49 ppm ^§^)	>60 h	Healthy	Early & mid-lactation; Milked 2x/day; 1 syringe per gland	[60] 1984
Penicillin	Sheep; NS; NS; *n* = 2	US Tol: Zero. EMA MRL: Not established.	LC-MS	0.005 ppm	NS	IM	1500 mg Benzylpenicillin daily	3	Liver	2 days (0.24 ppm)	>2 days	NS	NS	[56] 1996
									Kidney	2 days (0.87 ppm)	>2 days			
									Muscle	2 days (0.02 ppm)	>2 days			
Penicillin	Sheep; Lacaune; adult; *n* = 8	US Tol: Zero. EMA MRL: Not established.	Bioassay	0.006 ppm	NS	IMM	300,000 IU Procaine benzylpenicillin; 100 mg dihydro-streptomycin; 100 mg nafcillin	1	Milk	3 days (0.01 ppm ^§^)	4 days	Healthy	Dry off period (mean 112 days (range 85–223 days); 1 tube per gland before drying off. Sample collected after lambing	[28] 1995
Penicillin	Sheep; NS; 14–17 months; *n* = 10 study; *n* = 10/time pt	US Tol: Zero. EMA MRL: Not established.	Bioassay	0.0125 ppm	NS	IM	3000 IU/lb Penicillin G procaine daily	4	Liver	NS	9 days	Healthy	NS	[61] 2010
									Kidney	NS	9 days			
									Muscle	NS	9 days			
									Fat	NS	9 days			
									Inj. Site	NS	9 days			
Penicillin	Sheep; Awassi; adult; *n* = 3	US Tol: Zero. EMA MRL: Not established.	Bioassay	NS	NS	IV	20 mg/kg Penicillin ^†^, then 10 mg/kg for 4 doses 45 min interval	5	Milk	36 h (0.01 ppm ^§^)	48 h	Healthy	Lactating; Milked 2×/day	[24] 1973
			Radioactivity	NS	NS	IV	20 mg/kg Penicillin ^†^ (radiolabeled) then 10 mg/kg for 4 doses 45 min interval	5	Milk	8 h (0.08 ppm ^§^)	10 h			
Penicillin	Sheep; Sardinian; Adult; *n* = 5	US Tol: Zero. EMA MRL: Not established.	HPLC	2.6 ppb	8.8 ppb	IM	24 mg/kg Penicillin G sodium	1	Milk	8 days (0.01 ppm)	> 8 days	NS	Lactating; Milked 2×/day	[55] 1998
						IMM	24 mg/kg Penicillin G sodium	1	Milk	7 days (0.001 ppm)	8 days			
Penicillin	Sheep; domestic dairy breed; adult; *n* = 10	US Tol: Zero. EMA MRL: Not established.	Bioassay	3 ppb	4 ppb	IMM co-admin w/IM	1,000,000 IU Benzylpenicillin (IMM) daily co- administered with 250,000 IU benzylpenicillin (IM) at 24 h intervals.	5 (IMM) 2 (IM)	Milk	192 h (9.9 ppb)	>192 h	Diseased	Lactating; 1 syringe/gland	[52] 2009

^†^ Salt form unclear or not stated in article. # = number. * Projected time for which residues could still be detected based on study protocol for sample collection time points and sample concentration results. Authors caution readers to critically evaluate these publications to estimate when full residue depletion might occur. ^§^ Data points manually extracted use scanning software (Webplot digitizer or UnScanIt 7.0). Abbreviations: 2×/day: twice daily. LOD: Limit of detection. LOQ: Limit of quantification. EMA: European Medicines Agency. MRL: Maximum residue limit. ND: Not detected. NS: Not specified. Routes of Administration: IMM = intramammary, IM = intramuscular, IV = intravenous, PO = per os, POMF = per os as medicated feed, SC = subcutaneous. Units: s = seconds, min = minutes, h = hours, ppb = parts per billion, ppm = parts per million, mL = milliliter.

**Table 4 animals-12-02607-t004:** Cephalosporin residues in milk or edible tissue samples from sheep or goats following treatment.

Analyte	Species; Breed; Age; # of Animals	Tolerance/MRL	Analytical Method	LOD	LOQ	Route of Admini-stration	Dose & Active Ingredient	# of Doses	Matrix	Last Sampling Time Point (Post-Last Treatment) When Residues WERE Detected	Sampling Time Point When NO Residues Were Detected (Post-Last Treatment) *	Health Status	Additional Information	Source/Year
Cefepime	Goat; NS; Adult; *n* = 10	US Tol: Not established. EMA MRL: Not established.	Bioassay	NS	NS	IV	20 mg/kg Cefepime	1	Milk	12 h (0.17 ppm)	>12 h	Healthy	Lactating; Milked 2×/day	[62] 2004
						IM	20 mg/kg Cefepime	1	Milk	12 h (0.25 ppm)	>12 h			
Cefepime	Goat; NS; 1 year; *n* = 5	US Tol: Not established. EMA MRL: Not established.	HPLC	1.15 ppb	3.49 ppm	IM	50 mg/kg Cefepime	1	Milk	4 h (5.14 ppm^§^)	> 4 h	Healthy	Lactating	[72] 2010
Cefonicid	Goat; Muriano-Granadina; 2–4 years; *n* = 6	US Tol: Not established. EMA MRL: Not established.	HPLC	500 ppb	750 ppb	IV	10 mg/kg Cefonicid sodium	1	Milk	<LOQ @ 1 h	1 h	Healthy	Lactating; Milked 1×/day	[63] 2020
						IM	10 mg/kg Cefonicid sodium	1	Milk	<LOQ @ 1 h	1 h			
						SC	10 mg/kg Cefonicid sodium	1	Milk	<LOQ @ 1 h	1 h			
						SC	20 mg/kg Cefonicid sodium	1	Milk	<LOQ @ 1 h	1 h			
Cef-quinome	Goat; Zaraibi; 30–36 months; *n* = 5	US Tol: Not established. EMA MRL: Not established.	Bioassay	0.009 ppm	0.027 ppm	IV	3 mg/kg Cefquinome sulfate	1	Milk	48 h (0.02 ppm ^§^)	>2 days	Healthy	Lactating; Milked 1×/day	[67] 2015
			HPLC	0.006 ppm	0.017 ppm				Milk	48 h (0.01 ppm ^§^)	>2 days			
			Bioassay	0.009 ppm	0.027 ppm	IV	3 mg/kg Cefquinome sulfate	1	Milk	48 h (0.02 ppm ^§^)	>2 days	Diseased-Mastitis		
			HPLC	0.006 ppm	0.017 ppm				Milk	48 h (0.02 ppm ^§^)	>2 days			
Cef-quinome	Goat; Zaraibi; 30–36 months; *n* = 5	US Tol: Not established. EMA MRL: Not established.	HPLC	0.006 ppm	0.018 ppm	IMM	75 mg Cefquinome sulfate	1	Milk	120 h (0.01 ppm ^§^)	>120 h	Healthy	Early & mid-lactating; 1 full tube per gland 1 full tube into single infected udder half	[68] 2019
						IMM	75 mg Cefquinome sulfate	1	Milk	96 h (0.01 ppm ^§^)	120 h	Diseased-Mastitis		
Cef-tazidime	Goat; Creole; Adult; *n* = 6	US Tol: Not established. EMA MRL: Not established.	Bioassay	0.125 ppm	0.3 ppm	IV	10 mg/kg Ceftazidime	1	Milk	12 h (0.52 ppm ^§^)	>12 h	Healthy	Lactating; Milked 2x/day	[73] 2011
						IM	10 mg/kg Ceftazidime	1	Milk	12 h (0.54 ppm ^§^)	>12 h			
Ceftiofur	Goat; Alpine & Alpine-Saanen; 4 years; *n* = 6	US Tol: 100 ppb (milk). EMA MRL extrapolated from bovine to all mammalian species: 100 ppb (milk).	HPLC	NS	0.05 ppm	IV	2.2 mg/kg Ceftiofur sodium	1	Milk	24 h (NS)	2 days	Healthy	Lactating; Milked 2×/day	[71] 1994
						IM	2.2 mg/kg Ceftiofur sodium daily	5	Milk	24 h (NS)	2 days			
Ceftiofur	Sheep; NS; Adult; *n* = 9	US Tol: 100 ppb (milk). EMA MRL by extension from bovine to ovine: 100 ppb (milk).	HPLC	NS	NS	IM	2 mg/kg Ceftiofur sodium daily	5	Milk	<LOQ @ 12 h	12 h	Healthy	Lactating	[74] 2006
Ceftiofur	Goat; mixed dairy type; 28 months; *n* = 5	US Tol: 100 ppb (milk). EMA MRL extrapolated from bovine to all mammalian species: 100 ppb (milk).	LC-MS	NS	20 ppb	IMM	125 mg Ceftiofur hydrochloride daily	2	Milk	72 h (37 ppb)	4 days	Healthy	Mid- & late lactation; Milked 2×/day. Left udder half infused.	[75] 2015
Ceftriaxone	Goat; Dairy type; 1.5–2 years; *n* = 6	US Tol: Not established. EMA MRL: Not established.	Bioassay	NS	NS	IV	20 mg/kg Ceftriaxone sodium	1	Milk	2 h (0.11 ppm)	2.5 h	Healthy	Lactating	[64] 2013
Ceftriaxone	Goat; NS; 2–2.5 years; *n* = 10	US Tol: Not established. EMA MRL: Not established.	Bioassay	NS	0.2 ppm	IV	20 mg/kg Ceftriaxone	1	Milk	8 h (0.36 ppm)	10 h	Healthy	Lactating	[65] 2005
						IM	20 mg/kg Ceftriaxone	1	Milk	10 h (0.26 ppm)	12 h			
Ceftriaxone	Sheep; native breed; 2–3 years; *n* = 6	US Tol: Not established. EMA MRL: Not established.	Bioassay	NS	0.1 ppm	IV	10 mg/kg Ceftriaxone	1	Milk	10 h (0.22 ppm)	12 h	Healthy	Lactating; Milked 2×/day	[76] 2006
						IM	10 mg/kg Ceftriaxone	1	Milk	12 h (0.19 (ppm)	24 h			
Ceph-acetrile	Sheep; Texel; adult; *n* = 6	US Tol: Not established. EMA MRL: Not established.	Bioassay	NS	NS	IM	12 mg/kg Benzathine cephacetrile	1	Milk	24 h (NS)	>1 day	Healthy	Lactating	[66] 1977
Cephalexin	Goat; NS; 1 year; *n* = 2	US Tol: Not established. EMA MRL: Not established.	HPLC	0.165 ppm	NS	IM	10 mg/kg Cephalexin	1	Milk	72 h (0.07 ppm ^§^)	>3 days	NS	Lactating	[77] 2019
Cephalexin	Sheep; Awassi; adult; *n* = 10	US Tol: Not established. EMA MRL: Not established.	Bioassay	0.1 ppm	NS	IM	10 mg/kg Cephalexin	1	Milk	8 h (0.46 ppm ^§^)	>8 h	Healthy	Late lactation	[69] 1988
Ceph-alothin	Goat; Creole; adult; *n* = 20	US Tol: Not established. EMA MRL: Not established.	HPLC	0.01 ppm	NS	IV	10 mg/kg Cephalothin	1	Milk	12 h (0.31 ppm ^§^)	>12 h	Healthy	Lactating; Milked 2x/day	[78] 2004
Ceph-alothin	Goat; Creole; adult; *n* = 22 study; groups of 8, 8 and 6	US Tol: Not established. EMA MRL: Not established.	HPLC	0.01 ppm	NS	IV	20 mg/kg Cephalothin	1	Milk	6 h (0.08 ppm ^§^)	8 h	Healthy	Early lactation; Restricted diet	[79] 2007
						IV	20 mg/kg Cephalothin	1	Milk	8 h (0.28 ppm ^§^)	10 h		Early lactation; Restricted diet + additional energy	
						IV	20 mg/kg Cephalothin	1	Milk	12 h (0.12 ppm ^§^)	14 h		Early lactation; Balanced diet	
Cephapirin	Goat; French Alpine; 1–7 years; *n* = 20	US Tol: Not established. EMA MRL: Not established.	Bioassay	NS	NS	IMM	200 mg Cephapirin at 12 h intervals	2	Milk	ND @ 192 h	8 days	Healthy	Mid-lactation; 1 full tube into R half udder	[70] 1986
						IMM	200 mg Cephapirin at 12 h intervals	3	Milk	ND @ 192 h	8 days			
Cephapirin	Goat; dairy type; 2–7 years; *n* = 10	US Tol: Not established. EMA MRL: Not established.	Bioassay	NS	NS	IMM	200 mg Sodium cephapirin at 12 h intervals	2	Milk	48 h (0.03 ppm ^§^)	60 h	Healthy	Early & mid- lactation; Milked 2×/day; 1 full tube into each gland	[60] 1984
Cephradine	Goat; NS; adult; *n* = 4	US Tol: Not established. EMA MRL: Not established.	Spectrophoto-metrically	0.2 ppm	NS	IM	10 mg/kg Cephradine	1	Milk	8 h (1.55 ppm)	12 h	Healthy and Diseased	Lactating	[80] 1994
						IM	10 mg/kg Cephradine at 12 h intervals	3	Milk	8 h (1.28 ppm)	12 h			
						IM	10 mg/kg Cephradine at 12 h intervals	5	Milk	8 h (3.02 ppm)	12 h			
						IM	10 mg/kg Cephradine at 12 h intervals	7	Milk	8 h (2.78 ppm)	12 h			
						IM	10 mg/kg Cephradine at 12 h intervals	9	Milk	8 h (3.02 ppm)	12 h			

^§^ Data points manually extracted use scanning software (Webplot digitizer or UnScanIt 7.0). # = number. * Projected time for which residues could still be detected based on study protocol for sample collection time points and sample concentration results. Authors caution readers to critically evaluate these publications to estimate when full residue depletion might occur. Abbreviations: 1×/day: once daily. 2×/day: twice daily. LOD: Limit of detection. LOQ: Limit of quantification. EMA: European Medicines Agency. MRL: Maximum residue limit. ND: Not detected. NS: Not specified. Routes of Administration: IMM = intramammary, IM = intramuscular, IV = intravenous, SC = subcutaneous. Units: s = seconds, min = minutes, h = hours, ppb = parts per billion, ppm = parts per million.

**Table 5 animals-12-02607-t005:** Fluoroquinolone residues in milk or edible tissue samples from sheep or goats following treatment.

Analyte	Species; Breed; Age; # of Animals	Tolerance/ MRL	Analytical Method	LOD	LOQ	Route of Admini-stration	Dose & Active Ingredient	# of Doses	Matrix	Last Sampling Time Point (Post-Last Treatment) When Residues WERE Detected	Sampling Time Point When NO Residues Were Detected (Post-Last Treatment) *	Health Status	Additional Information	Source/ Year
Ciprofloxacin	Goats; NS; adult; *n* = 6	US Tol: Prohibited. EMA MRL: Not established.	Bioassay	0.05 ppm	NS	IV	4 mg/kg Ciprofloxacin	1	Milk	24 h (0.07 ppm)	30 h	Healthy	Lactating	[81] 2014
Ciprofloxacin	Goats; Baladi; 30–36 months; *n* = 5	US Tol: Prohibited. EMA MRL: Not established.	Bioassay	0.01 ppm	NS	IV	5 mg/kg Ciprofloxacin	1	Milk	10 h (0.11 ppm)	18 h	Healthy	Lactating	[82] 1998
						IM	5 mg/kg Ciprofloxacin	1	Milk	10 h (0.07 ppm)	18 h			
						IM	5 mg/kg Ciprofloxacin daily	5	Milk	3 days (0.07 ppm)	4 days			
Danofloxacin	Goats; Murciano-Granadina; 1.5–3 years; *n* = 6	US Tol: Prohibited. EMA MRL: Not established.	HPLC	0.005 ppm	0.015 ppm	SC	6 mg/kg Danofloxacin	1	Milk	36 h (0.01 ppm ^§^)	48 h	Healthy	Mid-lactation; Milked 2×/day	[83] 2007
Danofloxacin	Sheep; Manchega; 2–4 years; *n* = 6	US Tol: Prohibited. EMA MRL: Not established.	HPLC	0.005 ppm	0.015 ppm	SC	6 mg/kg Danofloxacin	1	Milk	36 h (0.02 ppm ^§^)	48 h	Healthy	Mid-lactation; Milked 2×/day	[83] 2007
Danofloxacin	Sheep; Assaf; adult; *n* = 12	US Tol: Prohibited. EMA MRL: Not established.	Bioassay	0.04 ppm	NS	IV	1.25 mg/kg Danofloxacin	1	Milk	24 h (0.1 ppm ^§^)	>1 day	Healthy	Mid-lactation	[99] 1997
						IM	1.25 mg/kg Danofloxacin	1	Milk	24 h (0.07 ppm ^§^)	>1 day			
Danofloxacin	Sheep; Assaf; 2–3 years; *n* = 5	US Tol: Prohibited. EMA MRL: Not established.	HPLC	4 ppb	5 ppb	IM	1.25 mg/kg	1	Milk	24 h (0.07 ppm ^§^)	>24 h	Healthy	Mid-lactation; Milked 2×/day	[96] 2011
						IM	1.25 mg/kg co-administered with 0.2 mg/kg ivermectin	1	Milk	24 h (0.09 ppm ^§^)	>24 h			
Danofloxacin	Sheep; Assaf; 2–3 years; *n* = 6	US Tol: Prohibited. EMA MRL: Not established.	HPLC	4 ppb	5 ppb	IM	1.25 mg/kg Danofloxacin	1	Milk	24 h (0.08 ppm ^§^)	>24 h	Healthy	Mid-lactation; Milked 2×/day	[97] 2013
						IM	1.25 mg/kg Danofloxacin + soy diet	1	Milk	24 h (0.1 ppm ^§^)	>24 h			
						IM	1.25 mg/kg Danofloxacin + Gen-daid (isoflavones)	1	Milk	24 h (0.03 ppm ^§^)	>24 h			
Danofloxacin	Sheep; Assaf; 2–3 years; *n* = 6	US Tol: Prohibited EMA MRL: Not established.	HPLC	NS	100 ppb	IM	1.25 mg/kg Danofloxacin	1	Milk	24 h (0.03 ppm ^§^)	>24 h	Healthy	Mid-lactation; Milked 2×/day	[100] 2013
						IM	1.25 mg/kg Danofloxacin co-administered with 1 mg/kg IV triclabendazole	1	Milk	24 h (0.25 ppm ^§^)	>24 h			
Danofloxacin	Sheep; Assaf; adult; *n* = 6	US Tol: Prohibited. EMA MRL: Not established.	HPLC	NS	19 ppb	IM	1.25 mg/kg Danofloxacin standard diet	1	Milk	24 h (0.05 ppm ^§^)	> 24 h	Healthy	Mid-lactation; Milked 2×/day	[101] 2018
						IM	1.25 mg/kg Danofloxacin w/ 10% flaxseed diet	1	Milk	24 h (0.04 ppm ^§^)	>24 hr			
						IM	1.25 mg/kg Danofloxacin w/ 15% flaxseed diet	1	Milk	24 h (0.05 ppm ^§^)	> 24 h			
Difloxacin	Goats; Murciano-Granadina; 4–5 years; *n* = 6	US Tol: Prohibited. EMA MRL: Not established.	HPLC	NS	10 ppb	IV	5 mg/kg Difloxacin	1	Milk	48 h (0.02 ppm ^§^)	72 h	Healthy	Lactating; Milked 1×/day	[102] 2010
						SC	5 mg/kg Difloxacin	1	Milk	36 h (0.02 ppm ^§^)	48 h			
						SC	15 mg/kg Difloxacin (long acting)	1	Milk	144 h (0.59 ppm ^§^)	>144 h			
Difloxacin	Goats; Murciano-Granadina; 4–5 years; *n* = 6	US Tol: Prohibited. EMA MRL: Not established	HPLC	NS	10 ppb	SC	15 mg/kg Difloxacin (long acting)	1	Milk	144 h (0.07 ppm ^§^)	>144 h	Healthy	Lactating; Milked 1x/day	[84] 2011
Enrofloxacin	Goats; Sham; 2–3 years; *n* = 10	US Tol: Prohibited. EMA MRL extension from bovine to all food producing species: 100 ppb (milk).	Bioassay	NS	0.02 ppm	IV	5 mg/kg Enrofloxacin	1	Milk	24 h (0.06 ppm)	36 h	Healthy	Mid-lactation; Milked 2x/day	[85] 2003
						IV	5 mg/kg Enrofloxacin co-administered with 7.5 mg/kg albendazole PO	1	Milk	12 h (0.11 ppm)	24 h			
						IM	5 mg/kg Enrofloxacin	1	Milk	36 h (0.08 ppm)	48 h			
						IM	5 mg/kg Enrofloxacin co-administered with 7.5 mg/kg albendazole PO	1	Milk	24 h (0.16 ppm)	36 h			
Enrofloxacin	Goats; Murciano-Granadina; 2.5–3.5 years; *n* = 6	US Tol: Prohibited. EMA MRL extension from bovine to all food producing species: 100 ppb (milk).	HPLC	NS	NS	SC	5 mg/kg Enrofloxacin	1	Milk	NS	NS ^†^	Healthy	Lactating	[103] 2009
Ciprofloxacin														
Enrofloxacin	Goats; Murciano-Granadina; 2.5–3.5 years; *n* = 6	US Tol: Prohibited. EMA MRL extension from bovine to all food producing species: 100 ppb (milk).	HPLC	NS	NS	IV	5 mg/kg Enrofloxacin	1	Milk	NS	NS ^‡^	Healthy	Lactating	[86] 2009
						SC	5 mg/kg Enrofloxacin (long acting)	1	Milk	NS	NS ^Ϙ^			
Enrofloxacin	Goats; NS; 1.5–2 years; *n* = 6	US Tol: Prohibited. EMA MRL extension from bovine to all food producing species: 100 ppb (milk).	Bioassay	0.01 ppm	NS	SC	5 mg/kg Enrofloxacin	1	Milk	30 h (0.08 ppm)	36 h	Healthy	Lactating	[87] 2009
						SC	5 mg/kg Enrofloxacin SC, pretreated with 70 mg/kg probenecid PO	1	Milk	36 h (0.02 ppm)	48 h			
Enrofloxacin	Sheep; NS; Neo-natal	US Tol: Prohibited. EMA MRL by extension from bovine to ovine: 300 ppb (liver); 200 ppb (kidney); 100 ppb (muscle, fat).	HPLC	NS	10 ppb	PO	7.5 mg/kg Enrofloxacin	1	Liver	NS	Enro ^∼^: 16 days	Healthy	NS	[104] 1998
											Cipro ^≈^: 16 days			
									Kidney	NS	Enro ^∼^: 16 days			
											Cipro ^≈^: 16 days			
Ciprofloxacin									Muscle	NS	Enro ^∼^: 16 days			
											Cipro ^≈^: 16 days			
									Fat	NS	Enro ^∼^: 16 days			
											Cipro ^≈^: 16 days			
Enrofloxacin	Sheep; crossbred; 2–4 years; *n* = 6	US Tol: Prohibited. EMA MRL extension from bovine to all food producing species: 100 ppb (milk).	Bioassay	NS	0.018 ppm	IV	2.5 mg/kg Enrofloxacin	1	Milk	24 h (0.13 ppm ^§^)	>24 h	Healthy	Lactating; Milked 2×/day	[88] 2003
						IM	2.5 mg/kg Enrofloxacin	1	Milk	24 h (0.15 ppm ^§^)	>24 h			
Enrofloxacin	Sheep; Assaf; 2–3 years; *n* = 12	US Tol: Prohibited. EMA MRL extension from bovine to all food producing species: 100 ppb (milk).	HPLC	NS	NS	IV	2.5 mg/kg Enrofloxacin	1	Milk	24 h (0.09 ppm ^§^)	> 24 h	Healthy	Mid-lactation; Milked 2×/day	[98] 2006
						IV	2.5 mg/kg Enrofloxacin co-administered with 0.8 mg/kg genistein IM	1	Milk	24 h (0.05 ppm ^§^)	> 24 h			
						IV	2.5 mg/kg Enrofloxacin co-administered with 2 mg/kg albendazole IV	1	Milk	24 h (0.06 ppm ^§^)	> 24 h			
Ibafloxacin	Goats; Murciano-Granadina; 3–4 years; *n* = 6	US Tol: Prohibited. EMA MRL: Not established.	HPLC	NS	10 ppb	IV	15 mg/kg Ibafloxacin	1	Milk	6 h (0.05 ppm ^§^)	12 h	Healthy	Lactating	[89] 2007
Levofloxacin	Goats; NS; 3–5 years; *n* = 6	US Tol: Prohibited. EMA MRL: Not established.	Bioassay	NS	0.05 ppm	IV	4 mg/kg Levofloxacin hemihydrate	1	Milk	36 h (0.04 ppm ^§^)	48 h	Healthy	Lactating	[90] 2009
						IM	4 mg/kg Levofloxacin hemihydrate	1	Milk	36 h (0.06 ppm ^§^)	48 h			
Marbo-floxacin	Goats; Anglo-nubian; 3–5 years; *n* = 6	US Tol: Prohibited. EMA MRL: Not established.	HPLC	NS	0.025 ppm	IV	5 mg/kg Marbofloxacin	1	Milk	36 h (0.06 ppm ^§^)	48 h	Healthy	Lactating	[91] 2017
						IM	5 mg/kg Marbofloxacin	1	Milk	36 h (0.07 ppm ^§^)	48 h			
Marbo-floxacin	Sheep; Assaf; adult; *n* = 15	US Tol: Prohibited. EMA MRL: Not established.	Bioassay	0.05 ppm ^	0.04 ppm ^	IV	2.5 mg/kg Marbofloxacin	1	Milk	24 h (0.05 ppm ^§^)	>24 h	Healthy	Mid-lactation	[92] 1997
						IM	2.5 mg/kg Marbofloxacin	1	Milk	24 h (0.23 ppm ^§^)	> 24 h			
Moxifloxacin	Goats; Murciano-Granadina; 3–4 years; *n* = 6	US Tol: Prohibited. EMA MRL: Not established.	HPLC	NS	10 ppb	IV	5 mg/kg Moxifloxacin	1	Milk	32 h (0.01 ppm ^§^)	48 h	Healthy	Lactating	[93] 2006
						SC	5 mg/kg Moxifloxacin	1	Milk	32 h (0.05 ppm ^§^)	48 h			
Moxifloxacin	Goats; Murciano-Granadina; 3–4 years; *n* = 6	US Tol: Prohibited. EMA MRL: Not established.	HPLC	NS	10 ppb	IM	5 mg/kg Moxifloxacin	1	Milk	32 h (0.01 ppm ^§^)	48 h	Healthy	Lactating	[105] 2007
Norfloxacin	Sheep; crossbred; adult; *n* = 6	US Tol: Prohibited. EMA MRL: Not established.	HPLC	0.07 ppm	NS	IV	25 mg/kg Norfloxacin nicotinate	1	Milk	24 h (10 ppm)	>24 h	Healthy	Lactating	[106] 1994
Orbifloxacin	Goats; Murciano-Granadina; 5–6 years; *n* = 6	US Tol: Prohibited. EMA MRL: Not established.	HPLC	20 ppb	25 ppb	IV	2.5 mg/kg Orbifloxacin	1	Milk	12 h (0.04 ppm ^§^)	24 h	Healthy	Lactating	[107] 2007
						SC	2.5 mg/kg Orbifloxacin	1	Milk	24 h (0.03 ppm ^§^)	36 h			
						IM	2.5 mg/kg Orbifloxacin	1	Milk	12 h (0.05 ppm ^§^)	24 h			
Orbifloxacin	Sheep; Barky; 4–6 years; *n* = 6	US Tol: Prohibited. EMA MRL: Not established.	Bioassay	NS	0.04 ppm	IV	2.5 mg/kg Orbifloxacin	1	Milk	24 h (0.09 ppm ^§^)	30 h	Healthy	Lactating	[94] 2009
						IM	2.5 mg/kg Orbifloxacin	1	Milk	30 h (0.06 ppm ^§^)	48 h			
Pefloxacin	Goats; Egyptian; 2 years; *n* = 5	US Tol: Prohibited. EMA MRL: Not established.	Bioassay	NS	0.078 ppm	IV	10 mg/kg Pefloxacin	1	Milk	10 h (0.1 ppm)	12 h	Healthy	Lactating	[95] 2002
						IM	10 mg/kg Pefloxacin	1	Milk	10 h (0.1 ppm)	12 h			
Flumequine	Sheep	US Tol: Not established. EMA established MRL: 100 ppb (liver); 300 ppb (kidney); 50 ppb (muscle, fat, skin).	HPLC	NS	100 ppb	IM	12 mg/kg Flumequine for first dose, then 6 mg/kg at 12 h intervals	10	Liver	Flu: 78 h (13.8 ppb)	Flu: >78 h	NS	NS	[108] 1997
										7-OH: 48 h(10.24 ppb)	7-OH: 60 h			
									Kidney	Flu: 78 h (38.6 ppb)	Flu: >78 h			
										7-OH: 78 h (4.5 ppb)	7-OH: >78 h			
									Muscle	Flu: 78 h (9.0 ppb)	Flu: >78 h			
7-Hydroxy-flumequine										7-OH: 18 h (15.3 ppb)	7-OH: 30 h			
									Fat	Flu: 78 h (52.5 ppb)	Flu: >78 h			
										7-OH: ND @ 18 h	7-OH: 18 h			
									Inj. Site	Flu: 90 h (10 ppb)	Flu: >90 h			
										7-OH: 30 h (13.5 ppb)	7-OH: 42 h			
Flumequine	Sheep; NS; NS; *n* = 20 study; *n* = 4/time pt	US Tol: Not established. EMA established MRL: 100 ppb (liver); 300 ppb (kidney); 50 ppb (muscle, fat, skin).	HPLC	NS	5 ppb	IM	12 mg/kg Flumequine for first dose, then 6 mg/kg at 12 h intervals	6	Liver	78 h (19.3 ppb)	>78 h	Healthy	NS	[109] 1998
									Kidney	78 h (62.5 ppb)	>78 h			
									Muscle	78 h (12.4 ppb)	>78 h			
									Fat	78 h (171.9 ppb)	>78 h			

^§^ Data points manually extracted use scanning software (Webplot digitizer or UnScanIt 7.0). # = number. ^†^ Enrofloxacin parent half-life reported = 2.74 h; Ciprofloxacin metabolite half-life = 4.79 h. ^‡^ Intravenous half-life reported= 5.39 h. ^Ϙ^ Subcutaneous half-life reported= 14.85 h. ^∼^ Enro: Enrofloxacin. ^≈^ Cipro: Ciprofloxacin. * Projected time for which residues could still be detected based on study protocol for sample collection time points and sample concentration results. Authors caution readers to critically evaluate these publications to estimate when full residue depletion might occur. ^ LOD and LOQ values should be confirmed with authors; however, they are reported as published. Abbreviations: 1×/day: once daily. 2×/day: twice daily. 7-OH: 7-hydroxyflumequine. LOD: Limit of detection. LOQ: Limit of quantification. EMA: European Medicines Agency. FLU: flumequine. MRL: Maximum residue limit. NS: Not specified. Routes of Administration: IM = intramuscular, IV = intravenous, PO = per os, SC = subcutaneous. Units: s = seconds, min = minutes, h = hours, ppb = parts per billion, ppm = parts per million.

**Table 6 animals-12-02607-t006:** Macrolide residues in milk or edible tissue samples from sheep or goats following treatment.

Analyte	Species; Breed Age; # of Animals	Tolerance/ MRL	Analytical Method	LOD	LOQ	Route of Admini-stration	Dose & Active Ingredient	# of Doses	Matrix	Last Sampling Time Point (Post-Last Treatment) When Residues WERE Detected	Sampling Time Point When NO Residues Were Detected (Post-Last Treatment) *	Health Status	Additional Information	Source/Year
Erythro-mycin	Goat; dairy type; 2–7 years; *n* = 10	US Tol: Not established. EMA MRL by extension from bovine to all food producing species: 40 ppb (milk).	Bioassay	NS	NS	IMM	300 mg Erythromycin at 12 h intervals	3	Milk	24 h (0.05 ppm ^§^)	36 h	Healthy	Early & mid-lactation; Milked 2x/day, Whole tube per gland	[60] 1984
Erythro-mycin	Goat; NS; adult; *n* = 6	US Tol: Not established. EMA MRL by extension from bovine to all food producing species: 40 ppb (milk).	Bioassay	NS	0.024 ppm	IV	10 mg/kg Erythromycin	1	Milk	12 h (0.14 ppm ^§^)	>12 h	Healthy	Early lactation	[121] 2007
						IM	15 mg/kg Erythromycin	1	Milk	12 h (0.24 ppm ^§^)	>12 h			
Erythro-mycin	Sheep; NS; 3–4 years; *n* = 6	US Tol: Not established. EMA MRL by extension from bovine to all food producing species: 40 ppb (milk).	Bioassay	NS	0.039 ppm	IV	10 mg/kg Erythromycin	1	Milk	12 h (0.14 ppm ^§^)	24 h	Healthy	Lactating	[111] 2007
						IM	10 mg/kg Erythromycin	1	Milk	12 h (0.16 ppm ^§^)	24 h			
						SC	10 mg/kg Erythromycin	1	Milk	24 h (0.05 ppm ^§^)	>24 h			
Erythro-mycin	Sheep; NS; NS; *n* = 20 study; *n* = 4/time pt	US Tol: Not established. EMA established MRL: 200 ppb (liver, kidney, muscle, fat).	Bioassay	NS	Liver: 250 ppb	IM	10 mg/kg Erythromycin daily	5	Liver	1 day (1.22 ppm)	3 days	Healthy	NS	[110] 2000
					Kidney: 250 ppb				Kidney	1 days (0.77 ppm)	3 days			
					Muscle: 200 ppb				Muscle	1 day (0.42 ppm)	3 days			
					Fat: 200 ppb				Fat	ND	1 day			
					Inj. Site: 200 ppb				Inj. Site	15 days (0.37 ppm)	>15 days			
			LS-MC	NS	100 ppb	IM	10 mg/kg Erythromycin daily	5	Liver	1 days (0.41 ppm)	3 days	Healthy		
									Kidney	1 days (0.59 ppm)	3 days			
									Muscle	1 days (0.27 ppm)	3 days			
									Fat	ND	1 day			
									Inj. Site	6 days (0.46 ppm)	9 days			
Gamithro-mycin	Sheep; Merino; 5–6 months; *n* = 9 study; *n* = 3/time pt	US Tol: Not established. EMA MRL: Not established.	LS-MC	NS	10 ppb	SC	6 mg/kg Gamithromycin	1	Skin	10 days (276 ppb)	>10 days	Healthy	NS	[125] 2014
Gamithro-mycin	Sheep; NS; 7 months; *n* = 35 study; *n* = 5/ time pt	US Tol: Not established. EMA established MRL: 300 ppb (liver); 200 ppb (kidney); 50 ppb (muscle & fat).	LS-MC	NS	NS	SC	6 mg/kg Gamithromycin	1	Liver	NS ^‡^	NS ^‡^	Healthy	NS	[126] 2016
									Kidney	NS ^‡^	NS ^‡^			
									Muscle	NS ^‡^	NS ^‡^			
									Fat	NS ^‡^	NS ^‡^			
									Inj. Site	NS ^‡^	NS ^‡^			
Spiramycin	Sheep; Awassi; adult; *n* = 1	US Tol: Not established. EMA MRL: Not established.	Bioassay	NS	NS	IV	20 mg/kg Spiramycin adipate	1	Milk	60 h (2.78 ppm ^§^)	>96 h	Healthy	Lactating; Milked 2×/day	[24] 1973
			Radio-activity	NS	NS	IV	20 mg/kg Spiramycin (radiolabeled)	1	Milk	60 h (3.61 ppm ^§^)	>96 h			
Spiramycin	Sheep; Awassi; Adult; *n* = 2	US Tol: Not established. EMA MRL: Not established.	Bioassay	NS	NS	IM	20 mg/kg Spiramycin adipate	1	Milk	56 h (2.40 ppm ^§^)	>56 h	Healthy	Lactating; Milked 2x/day	[29] 1974
			Radio-activity	NS	NS				Milk	56 h (3.61 ppm ^§^)	>56 h			
			Bioassay	NS	NS	IM	20 mg/kg Spiramycin (radiolabeled)	1	Milk	56 h (1.79 ppm ^§^)	>56 h	Diseased- mastitis		
			Radio-activity	NS	NS				Milk	56 h (1.92 ppm ^§^)	>56 h			
Tilmicosin	Goats; NS; 2.5–3 years; *n* = 5	US Tol: Not established. EMA MRL by extension from ovine to all food producing species: 40 ppb (milk).	Bioassay	10 ppb	NS	SC	10 mg/kg Tilmicosin	1	Milk	11 days (0.16 ppm ^§^)	12 days	Healthy	Early lactation	[112] 1997
Tilmicosin	Sheep; Barki; 2–3 years; *n* = 5	US Tol: Not established. EMA established MRL: 40 ppb (milk).	Bioassay	NS	0.1 ppm	SC	10 mg/kg Tilmicosin phosphate	1	Milk	8 days (0.04 ppm ^§^)	>8 days	Healthy	Mid-lactation	[127] 1999
Tilmicosin	Sheep; Suffolk crossbred; adult; *n* = 4	US Tol: Not established. EMA established MRL: 40 ppb (milk).	HPLC	50 ppb	NS	SC	10 mg/kg Tilmicosin	1	Milk	11 days (46 ppb)	>11 days	NS	Early & mid-lactation; Milked 2x/day	[128] 1994
Tilmicosin	Sheep; Beulah Cross; 10–11 weeks; *n* = 14 study (slaughter 4 time pts)	US Tol: 1200 ppb (liver); 100 ppb (muscle). EMA established MRL: 1000 ppb (liver & kidney); 50 ppb (muscle & fat).	Radio-activity	NS	NS	SC	20 mg/kg Tilmicosin phosphate (radiolabeled)	1	Liver	28 days (2.7 ppm)	>28 days	Healthy	NS	[129] 2002
									Kidney	28 days (0.55 ppm)	>28 days			
									Muscle	28 days (1.35 ppm)	>28 days			
									Fat	28 days (0.26 ppm)	>28 days			
									Inj. Site	28 days (6.51 ppm)	>28 days			
	NS; NS; *n* = 16 & 4 slaughter time pts		HPLC	50 ppb	NS	SC	20 mg/kg Tilmicosin phosphate	1	Liver	28 days (160 ppb)	>28 days	Healthy		
									Kidney	28 days (73 ppb)	>28 days			
									Muscle	7 days (193.5 ppb)	21 days			
									Fat	3 days (73 ppb)	7 days			
									Inj. Site	28 days (121.8 ppb)	>28 days			
	Scottish Blackface; 6 months; *n* = 24 study; *n* = 4/time pt		HPLC	NS	50 ppb	SC	30 mg/kg Tilmicosin phosphate	1	Liver	56 days (81 ppb)	>56 days	Healthy		
									Kidney	42 days (51 ppb)	56 days			
									Muscle	<LOQ @ 14 days	14 days			
									Fat	<LOQ @ 14 days	14 days			
									Inj. Site	56 days (81 ppb)	>56 days			
	Swaledale; NS; *n* = 28 study; *n* = 4/tme pt		HPLC	NS	50 ppb	SC	10 mg/kg Tilmicosin phosphate	1	Liver	35 days (59 ppb)	42 days	Healthy		
									Kidney	21 days (73 ppb)	28 days			
									Muscle	<LOQ @ 14 days	14 days			
									Fat	<LOQ @ 14 days	14 days			
									Inj. Site	28 days (80 ppb)	35 days			
Tilmicosin	Sheep; NS; lambs; *n* = 12 study; *n* = 3/time pt	US Tol: 1200 ppb (liver); 100 ppb (muscle). EMA established MRL: 50 ppb (muscle & fat); 1000 ppb (liver & kidney).	Radio-activity	NS	NS	SC	20 mg/kg Tilmicosin phosphate (radiolabeled)	1	Liver	28 days (2.7 ppm)	>28 days	Healthy	NS	[113] 1997
									Kidney	28 days (0.55 ppm)	>28 days			
									Muscle	28 days(<0.26 ppm)	>28 days			
									Fat	28 days (<1.2 ppm)	>28 days			
	NS; lambs; *n* = 12 study; *n* = 3/time pt								Inj. Site	28 days (1.32 ppm)	>28 days			
			HPLC	NS	0.05 ppm	SC	20 mg/kg Tilmicosin phosphate	1	Liver	28 days (0.16 ppm)	>28 days	Healthy	NS	
									Kidney	28 days (0.06 ppm)	>28 days			
									Muscle	7 days (0.19 ppm)	21 days			
	Swaledale; NS; *n* = 28 study; *n* = 4/time pt								Fat	7 days (<0.05 ppm)	7 days			
									Inj. Site	28 days (0.12 ppm)	>28 days			
			LC	NS	0.05 ppm	SC	10 mg/kg Tilmicosin phosphate	1	Liver	21 days (0.07 ppm)	28 days	Healthy	NS	
									Kidney	21 days (0.07 ppm)	28 days			
	NS; adult; *n* = 4								Muscle	ND @ 14 days	14 days			
									Fat	<LOQ @ 14 days	14 days			
									Inj. Site	28 days (0.08 ppm)	35 days			
			HPLC	NS	0.05 ppm	SC	10 mg/kg Tilmicosin phosphate	1	Milk	10 days (0.06 ppm)	14 days	Healthy	Lactating	
Tilmicosin	Sheep; Suffolk crossbred; Adult; *n* = 4	US Tol: Not established. EMA established MRL: 50 ppb (milk).	HPLC	NS	50 ppb	SC	10 mg/kg Tilmicosin	1	Milk	15 days (0.3 ppm ^§^)	>15 days	Healthy	Early lactation	[114] 2008
Tulathro-mycin	Goats; Boer; 5–7 months; *n* = 16 study; *n* = 4/time pt	US Tol: Not established. EMA established MRL: 450 ppb (muscle); 250 ppb (fat); 5400 ppb (liver);1800 ppb (kidney).	UPLC	0.7 ppb	2 ppb	SC	2.5 mg/kg Tulathromycin at 7-day interval	2	Liver	20 days (0.78 ppm))	>20 days	Healthy	NS	[123] 2012
									Kidney	20 days (0.44 ppm)	>20 days			
									Muscle	5 days (0.46 ppm)	10 days			
									Fat	10 days (0.17 ppm)	20 days			
Tulathro-mycin	Goats; Mixed; 7–8 weeks; *n* = 6	US Tol: Not established. EMA established MRL: 450 ppb (muscle); 250 ppb (fat); 5400 ppb (liver); 1800 ppb (kidney).	LC-MS	Liver: 0.75 ppm	Liver: 1.91 ppm	SC	2.5 mg/kg Tulathromycin	1	Liver	<LOD @ 14 days	14 days	Healthy Juveniles	NS	[124] 2012
									Kidney	<LOD @ 14 days	14 days			
				Kidney: 0.29 ppm	Kidney: 1.66 ppm				Muscle	<<LOD @ 14 days	14 days			
									Fat	<LOD @ 14 days	14 days			
				Muscle: 0.24 ppm	Muscle: 0.69 ppm				Inj. Site	35 days (0.25 ppm)	>35 days			
	Mixed; 5–6 months; *n* = 30 stdy; *n* = 6/time pt					SC	2.5 mg/kg Tulathromycin	1	Liver	12 days (1.18 ppm)	18 days	Healthy Market-age		
				Fat: 0.14 ppm	Fat: 0.61 ppm				Kidney	48 days (0.31 ppm)	>48 days			
									Muscle	5 days (0.24 ppm)	12 days			
				Inj. Site: 0.24 ppm	Inj. Site: 0.69 ppm				Fat	12 days (0.15 ppm)	18 days			
									Inj. Site	18 days (1.27 ppm)	27 days			
	Mixed; 2–3 weeks; *n* = 12					SC	2.5 mg/kg Tulathromycin at 7-day interval	3	Liver	7 days (0.7 ppm)	>7 days	Healthy Juveniles		
									Kidney	<LOD @ 7 days	>7 days			
									Muscle	<LOD @ 7 days	>7 days			
									Fat	<LOD @ 7 days	>7 days			
									Inj. Site	7 days (8.76 ppm)	>7 days			
						SC	7.5 mg/kg Tulathromycin at 7-day interval	3	Liver	7 days (3.4 ppm)	>7 days	Healthy Juveniles		
									Kidney	7 days (1.65 ppm)	>7 days			
									Muscle	7 days (0.65 ppm)	>7 days			
									Fat	7 days (0.36 ppm)	>7 days			
									Inj. Site	7 days (17.9 ppm)	>7 days			
						SC	12.5 mg/kg Tulathromycin at 7-day interval	3	Liver	7 days (4.87 ppm)	>7 days	Healthy Juveniles		
									Kidney	7 days (3.28 ppm)	>7 days			
									Muscle	7 days (1.33 ppm)	>7 days			
									Fat	7 days (0.65 ppm)	>7 days			
									Inj. Site	7 days (24.4 ppm)	>7 days			
Tulathro-mycin	Goats; dairy type; 2–5 years; *n* = 8	US Tol: Not established. EMA MRL: Not established.	HPLC	1.8 ppb	5.0 ppb	SC	2.5 mg/kg Tulathromycin	1	Milk	45 days (2 ppb)	>45 days	Healthy,	Lactating; Milked 2×/day	[115] 2016
Tulathro-mycin	Goats; NS; 30–36 months; *n* = 5	US Tol: Not established. EMA MRL: Not established.	Bioassay	NS	NS	IV	2.5 mg/kg Tulathromycin	1	Milk	19 days (0.08 ppm ^§^)	>19 days	Healthy	Lactating	[116] 2012
						IM	2.5 mg/kg Tulathromycin	1	Milk	19 days (0.1 ppm ^§^)	>19 days			
Tulathro-mycin	Goats; dairy type; 1–7 years; *n* = 8	US Tol: Not established. EMA MRL: Not established.	LS-MS	1.8 ppb	5.0 ppb	SC	2.5 mg/kg Tulathromycin at 7-day interval	2	Milk	58 days (0.5 ppb)	61 days	Healthy	Lactating; Milked 2×/day	[117] 2016
Tulathro-mycin	Sheep; NS; NS; *n* = 30 study; *n* = 3/time pt	US Tol: Not established. EMA established MRL: 450 ppb (muscle); 250 ppb (fat); 5400 ppb (liver); 1800 ppb (kidney).	LS-MC	NS	Liver: 300 ppb	IM	2.5 mg/kg Tulathromycin	1	Liver	35 days (0.3 ppm)	42 days	Healthy	NS	[130] 2015
					Kidney: 200 ppb				Kidney	21 days (0.2 ppm)	28 days			
					Muscle; 50 ppb				Muscle	21 days (0.05 ppm)	28 days			
					Fat: 50 ppb				Fat	14 days (0.05 ppm)	21 days			
					Inj. Site: 50 ppb				Inj. Site	49 days (0.15 ppm)	>49 days			
Tylosin	Goats; NS; adult; *n* = 5	US Tol: Not established. EMA MRL by extension from bovine to all food producing species: 50 ppb (milk).	Bioassay	NS	NS	IV	15 mg/kg Tylosin tartrate	1	Milk	24 h (0.6 ppm)	>24 h	Healthy	Lactating	[118] 1991
						IM	15 mg/kg Tylosin tartrate	1	Milk	24 h (1.7 ppm)	>24 h			
Tylosin	Sheep; Awassi; adult; *n* = 3	US Tol: Not established. EMA MRL by extension from bovine to all food producing species: 50 ppb (milk).	Bioassay	NS	NS	IM	20 mg/kg Tylosin	1	Milk	26 h (1.80 ppm)	>26 h	Healthy	Lactating; Milked 2x/day	[119] 1973
									Milk	26 h (0.67 ppm)	>26 h	Diseased- mastitis		
Tylosin	Sheep; Merino; adult; *n* = 7	US Tol: Not established. EMA MRL by extension from bovine to all food producing species: 50 ppb (milk).	HPLC	NS	NS	IM	10 mg/kg Tylosin	5	Milk	36 h (30.9 ppb)	48 h	Healthy	Lactating; Milked 2×/day	[120] 2001

^§^ Data points manually extracted use scanning software (Webplot digitizer or UnScanIt 7.0). # = number. ^‡^ Liver HL reported= 5.48 days; Kidney HL reported = 4.22 days; Muscle HL = 2.55 days; Fat HL reported = 2.82 days; Injection site core = 4.43 days; Injection site ring = 2.39 days. * Projected time for which residues could still be detected based on study protocol for sample collection time points and sample concentration results. Authors caution readers to critically evaluate these publications to estimate when full residue depletion might occur. Abbreviations: 2×/day: twice daily. LOD: Limit of detection. LOQ: Limit of quantification. EMA: European Medicines Agency. MRL: Maximum residue limit. ND: Not detected. NS: Not specified. Routes of Administration: IMM = intramammary, IM = intramuscular, IV = intravenous, SC = subcutaneous. Units: s = seconds, min = minutes, h = hours, ppb = parts per billion, ppm = parts per million.

**Table 7 animals-12-02607-t007:** Sulfonamide residues in milk or edible tissue samples from sheep or goats following treatment.

Analyte	Species; Breed; Age; # of Animals	Tolerance/MRL	Analytical Method	LOD	LOQ	Route of Admini-stration	Dose & Active Ingredient	# of Doses	Matrix	Last Sampling Time Point (Post-Last Treatment) When Residues WERE Detected	Sampling Time Point When NO Residues Were Detected (Post-Last Treatment) *	Health Status	Additional Information	Source/Year
Sulfa-dimethoxine	Goat; NS; adult; *n* = 5	US Tol: Not established. EMA MRL: Not established.	Color-imetrically	NS	NS	PO	286 mg/kg sulfadimethoxine	1	Milk	2 days (NS)	3 days	Healthy	Lactating	[135] 2016
Sulfa-nilamide	Goat; NS; Adult; *n* = 1	US Tol: Not established. EMA established MRL: 100 ppb (milk).	Spectro-metric	NS	NS	IMM	1000 mg Sulfanilamide	1	Milk	4 days (143 ppm)	>4 days	Healthy	Lactating; Single gland	[132] 1958
Sulfa-cetamide	Goat; NS; Adult; *n* = 1	US Tol: Not established. EMA established MRL: 100 ppb (milk).	Spectro-metric	NS	NS	IMM	1000 mg Sulfacetamide	1	Milk	4 days (2520 ppm)	>4 days	Healthy	Lactating; Single gland	[132] 1958
Sulfa-nilamide	Sheep; NS; adult; *n* = 7	US Tol: Not established. EMA MRL: Not established.	Bioassay	NS	NS	IV; PO	150 mg/kg Sulfanilamide IV once then 100 mg/kg sulfanilamide PO at 12 h intervals	8	Liver	8 days (79 ppm)	>8 days	Healthy	NS	[136] 1977
									Kidney	8 days (119 ppm)	>8 days			
									Muscle	8 days (50 ppm)	>8 days			
Sulfa-methoxy-pyridazine	Sheep; NS; adult; *n* = 7	US Tol: Not established. EMA MRL: Not established.	Bioassay	NS	NS	PO	35 mg/kg Sulfamethoxy-pyridazine once then 25 mg/kg Sulfamethoxy-pyridazine daily	4	Liver	8 days (55 ppb)	>8 days	Healthy	NS	[136] 1977
									Kidney	8 days (115 ppb)	>8 days			
									Muscle	8 days (41 ppb)	>8 days			
Sulf-athiazole	Sheep; mixed; lambs; *n* = 15 study; *n* = 3/time pt	US Tol: Not established. EMA MRL: Not established.	Spectrometric	NS	NS	IV	72 mg/kg Sodium sulfathiazole	1	Liver	1 day (0.12 ppm ^§^)	>1 day	Healthy	NS	[137] 1977
									Kidney	1 days (0.11 ppm ^§^)	>1 day			
									Muscle	16 h (0.27 ppm ^§^)	1 day			
									Fat	16 h (0.26 ppm ^§^)	1 day			
Sulfa-merazine	Sheep; mixed; 22 months; *n* = 13 study; *n* = 3/time pt	US Tol: Not established. EMA MRL: Not established.	HPLC	NS	0.1 ppm	PO	132 mg/kg Sulfamerazine once, then 66 mg/kg at 12 h intervals	6	Liver	5 days (0.11 ppm)	7 days	Healthy	NS	[138] 1972
									Kidney	5 days (0.07 ppm)	7 days			
									Muscle	7 days (0.12 ppm)	10 days			
									Fat	7 days (0.05 ppm)	>7 days			
Sulfa-merazine	Sheep; NS; adult; *n* = 12	US Tol: Not established. EMA established MRL: 100 ppb (milk).	Spectro-metric	NS	NS	PO	100 mg/kg Sulfamerazine	1	Milk	2 days (3.7 ppm)	>2 days	Healthy	Lactating; Full dose/ gland	[139] 1978
						IV	100 mg/kg Sulfamerazine	1	Milk	1 day (5.0 ppm)	2 days			
						IM	100 mg/kg Sulfamerazine	1	Milk	1 days (4.2 ppm)	2 days			
						IMM	500 mg Sulfamerazine	1	Milk	12 min (428 ppm)	>12 min			
Sulfa-methazine ^†^	Goat; West African Dwarf; 1 year; *n* = 20 study; *n* = 1/time point	US Tol: Not established. EMA MRL: Not established.	Spectro-metric	0.05 ppm	NS	IM	100 mg/kg Sulfadimidine	1	Liver	30 days (5.29 ppm)	>30 days	Healthy	NS	[133] 2018
									Kidney	30 days (3.84 ppm)	>30 days			
									Muscle	30 days (2.01 ppm)	>30 days			
									Fat	30 days (4.84 ppm)	>30 days			
						IM	100 mg/kg Sulfadimidine co-admin w/5 mg/kg piroxicam	1	Liver	30 days (5.33 ppm)	>30 days			
									Kidney	30 days (4.79 ppm)	>30 days			
									Muscle	30 days (1.38 ppm)	>30 days			
									Fat	30 days (4.53 ppm)	>30 days			
Sulfa-methazine ^†^	Sheep; Targhee/Rambouillet; lambs; *n* = 16 study; *n* = 2/time pt	US Tol: Not established. EMA MRL: Not established.	Spectro-metric	NS	NS	IV	107.25 mg/kg Sodium sulfamethazine	1	Liver	4 days (0.11 ppm)	>4 days	Healthy	NS	[134] 1977
									Kidney	4 days (0.14 ppm)	>4 days			
									Muscle	4 days (0.09 ppm)	>4 days			
									Fat	4 days (0.05 ppm)	>4 days			
Sulfa-methazine ^†^	Sheep; crossbred; 2–3 years; *n* = 25 study; *n* = 5/time pt	US Tol: Not established. EMA MRL: Not established.	HPLC	NS	0.1 ppm	PO	391 mg/kg Sulfamethazine	1	Liver	4 days (0.3 ppm)	8 days	Healthy	NS	[140] 1991
									Kidney	4 days (0.25 ppm)	8 days			
									Muscle	4 days (0.2 ppm)	8 days			
									Fat	ND	4 days			
Sulfa-methazine ^†^	Sheep; crossbred; adult; *n* = 10	US Tol: Not established. EMA Established MRL: 100 ppb (milk).	Spectro-metric	NS	NS	PO	15,000 mg Sulfamethazine	1	Milk	1 day (NS)	>1 day	Healthy	Lactating	[131] 1965
						PO	15,000 mg Sulfamethazine at 12 h interval	2	Milk	2 days (NS)	>2 days	Healthy		
						PO	15,000 mg Sulfamethazine at 16 h interval	2	Milk	2 days (NS)	>2 days	Healthy		
						PO	15,000 mg Sulfamethazine at 22 h interval	2	Milk	2 days (NS)	>2 days	Healthy		
						PO	15,000 mg Sulfamethazine at 24 h interval	2	Milk	2 days (NS)	>2 days	Healthy		
						PO	15,000 mg Sulfamethazine at 25 h interval	2	Milk	53 h (NS)	>53 h	Diseased- mastitis		
						PO	15,000 mg Sulfamethazine first dose, 10,000 mg second dose at 24 h interval	2	Milk	2 days (NS)	>2 days	Healthy		
						PO	15,000 mg Sulfamethazine first dose, 7000 mg second dose at 24 h interval	3	Milk	2 days (NS)	>2 days	Healthy		
						PO	15,000 mg Sulfamethazine first dose, 7000 mg second dose at 22 h interval	3	Milk	78 h (NS)	>78 h	Diseased- mastitis		
						PO	15,000 mg Sulfamethazine first 2 doses at 13 h interval, 7000 mg third dose at 23 h interval	3	Milk	74 h (NS)	>74 h	Diseased- mastitis		
						PO	15,000 mg Sulfamethazine first 2 doses at 13 h interval, 7000 mg third dose at 22 h interval	3	Milk	83 h (NS)	>83 h	Diseased- mastitis		
						PO	18,000 mg first dose, 6000 mg second dose at 17 h interval then 19 h interval	3	Milk	80 h (NS)	>80 h	Diseased- mastitis		
						PO	18,000 mg Sulfamethazine first dose, 6000 mg at 24 h intervals	4	Milk	96 h (NS)	>96 h	Diseased- mastitis		
Sulfa-methazine ^†^	Sheep; NS; NS; NS	US Tol: Not established. EMA MRL: Not established.	NS	NS	NS	PO	107.25 mg/kg Sulfamethazine	1	Liver	2 days (0.1 ppm ^§^)	>2 days	Healthy	NS	[141] 1978
									Kidney	2 days (0.23 ppm ^§^)	>2 days			
									Muscle	2 days (0.15 ppm ^§^)	>2 days			
									Fat	36 h (0.16 ppm ^§^)	2 days			
Sulfa-methazine ^†^	Sheep; Suffolk; NS; *n* = 2; *n* = 1/time pt	US Tol: Not established. EMA MRL: Not established.	Radioactivity	NS	NS	PO	100 mg/kg Sulfamethazine (radiolabeled)	1	Liver	2 days (10 ppm)	>2 days	Healthy	NS	[142] 1983
									Kidney	2 days (22 ppm)	>2 days			
									Muscle	2 days (3 ppm)	>2 days			
Sulfa-methazine ^†^	Sheep; Balady; 2–4 years; *n* = 9 study; *n* = 3/time pt	US Tol: Not established. EMA MRL: Not established.	NS	NS	NS	IM	0.1 mg/kg Sulfadimidine	1	Liver	4 h (20 ppm)	>4 h	Healthy	NS	[143] 1980
									Kidney	4 h (198 ppm)	>4 h			
									Muscle	4 h (11 ppm)	>4 h			
Sulfadiazine	Sheep; Balady; 2–4 years; *n* = 9 study; *n* = 3/time pt	US Tol: Not established. EMA MRL: Not established.	NS	NS	NS	IM	0.1 mg/kg Sulfadiazine	1	Liver	4 h (25 ppm)	>4 h	Healthy	NS	[143] 1980
									Kidney	4 h (40 ppm)	>4 h			
									Muscle	4 h (13 ppm)	>4 h			

^§^ Data points manually extracted use scanning software (Webplot digitizer or UnScanIt 7.0). # Number. ^†^ Sulfamethazine and sulfadimidine are the same chemical/active ingredient. * Projected time for which residues could still be detected based on study protocol for sample collection time points and sample concentration results. Authors caution readers to critically evaluate these publications to estimate when full residue depletion might occur. Abbreviations: LOD: Limit of detection. LOQ: Limit of quantification. EMA: European Medicines Agency. MRL: Maximum residue limit. NS: Not specified. Routes of Administration: IMM = intramammary, IM = intramuscular, IV = intravenous, PO = per os. Units: s = seconds, min = minutes, h = hours, ppb = parts per billion, ppm = parts per million.

**Table 8 animals-12-02607-t008:** Tetracycline residues in milk or edible tissue samples from sheep or goats following treatment.

Analyte	Species;Breed; Age; # of Animals	Tolerance/MRL	Analytical Method	LOD	LOQ	Route of Admini-stration	Dose & Active Ingredient	# of Doses	Matrix	Last Sampling Time Point (Post-Last Treatmnet) When Residues WERE Detected	Sampling Time Point When NO Residues Were Detected (Post-Last Treatment) *	Health Status	Additional Information	Source/Year
Chlortetra-cycline	Sheep; Chios & Friesian; adult; *n* = 4	US Tol: Not established. EMA established MRL for all food producing species: 100 ppb (milk).	Bioassay	NS	NS	IM	25 mg/kg Chlor-tetracycline hydrochloride	1	Milk	72 h (0.1 ppm) 120 h (R udder) (0.28 ppm)	>72 h >120 h (R udder)	Healthy	Lactating; Only right ½ of udder infused.	[147] 1982
						IMM	426 mg Chlor-tetracycline hydrochloride in right half of udder.	1	Milk	38 h (L udder) (0.09 ppm)	48 h (L udder)			
Chlortetra-cycline	Sheep; NS; lambs; NS	US Tol: 6000 ppb (liver); 12,000 ppb (kidney, fat); 2000 ppb (muscle). EMA established MRL for all food producing species: 300 ppb (liver); 600 ppb (kidney); 100 ppb (muscle).	NS	NS	Liver: 0.03 ppm	POMF	50 mg/kg Chlor-tetracycline daily	42	Liver	0 days (0.11 ppm)	2 days	Healthy	NS	[148] 1996
					Kidney: 0.028 ppm				Kidney	2 days (0.06 ppm)	4 days			
					Muscle: 0.027 ppm				Muscle	0 days (0.03 ppm)	2 days			
					Fat: 0.025 ppm				Fat	ND	0 days			
Chlortetra-cycline	Sheep; NS; lambs; NS	US Tol: 6000 ppb (liver); 12,000 ppb (kidney, fat); 2000 ppb (muscle). EMA established MRL for all food producing species: 300 ppb (liver); 600 ppb (kidney); 100 ppb (muscle).	NS	NS	Liver: 0.03 ppm	POMF	50 mg/kg Chlor-tetracycline co-admin with 50 mg/kg sulfamethazine daily	42	Liver	0 days (0.21 ppm)	4 days	Healthy	NS	[148] 1996
					Kidney: 0.028 ppm				Kidney	6 days (0.05 ppm)	8 days			
					Muscle: 0.027 ppm				Muscle	0 days (0.04 ppm)	4 days			
					Fat: 0.025 ppm				Fat	ND	0 days			
Doxy-cycline	Goat; NS; adult; *n* = 6	US Tol: Not established. EMA MRL: Not established.	Bioassay	NS	NS	IV	5 mg/kg Doxycycline hydrochloride	1	Milk	48 h (0.12 ppm ^§^)	>2 days	Healthy	Lactating	[149] 1989
Mino-cycline	Goat; NS; 1.5–2 years; *n* = 6	US Tol: Not established. EMA MRL: Not established.	Bioassay	NS	NS	IV	5 mg/kg Minocycline hydrochloride	1	Milk	36 h (0.11 ppm)	2 days	Healthy	Lactating	[150] 1999
Oxytetra-cycline	Sheep; Chios & Friesian; adult; *n* = 4	US Tol: Not established. EMA established MRL for all food producing species: 100 ppb (milk).	Bioassay	NS	NS	IM	30 mg Oxytetracycline hydrochloride	1	Milk	38 h (0.7 ppm)	48 h	Healthy	Lactating; Only right ½ of udder infused.	[147] 1982
						IMM	420 mg Oxytetracycline hydrochloride in right half of udder	1	Milk	110 h (R udder) (0.58 ppm)	120 h (R udder)			
										14 h (L udder) (1.22 ppm	24 h (L udder)			
Oxytetra-cycline	Sheep; Awassi; adult; *n* = 8	US Tol: Not established. EMA established MRL for all food producing species: 100 ppb (milk).	Bioassay	0.5 ppm	NS	IM	20 mg/kg Oxytetracycline	1	Milk	72 h (NS)	>3 days	Healthy	Early lactation	[151] 1982
Oxytetra-cycline	Sheep; mixed breed; NS; *n* = 24 study; *n* = 4/time pt	US Tol: 6000 ppb (liver); 12,000 ppb (kidney, fat); 2000 ppb (muscle). EMA established MRL for all food producing species: 300 ppb (liver); 600 ppb (kidney); 100 ppb (muscle).	HPLC	NS	Liver: 85 ppb	IM	19.8 mg/kg Oxytetracycline (long acting)	1	Liver	NS	14 days	NS	NS	[152] 1997
					Kidney: 42 ppb				Kidney	NS	14 days			
					Muscle: 45 ppb				Muscle	NS	14 days			
					Fat: 45 ppb				Fat	NS	14 days			
									Inj. Site	NS	14 days			
Oxytetra-cycline	Sheep; Sardinian; adult; *n* = 5	US Tol: Not established. EMA established MRL for all food producing species: 100 ppb (milk).	HPLC	5.2 ppb	17.5 ppb	IMM	20 mg/kg Oxytetracycline	1	Milk	7 days (0.1 ppm ^§^)	>7 days	NS	Lactating; Milked 2×/day	[153] 1999
						IM	20 mg/kg Oxytetracycline	1	Milk	7 days (4.15 ppm ^§^)	>7 days			
Oxytetra-cycline	Sheep; mixed breed; NS; *n* = 24 study; *n* = 4/time pt	US Tol: 6000 ppb (liver);12,000 ppb (kidney, fat); 2000 ppb (muscle). EMA established MRL for all food producing species: 300 ppb (liver); 600 ppb (kidney); 100 ppb (muscle).	HPLC	NS	Liver: 85 ppb	IM	20 mg/kg Oxytetracycline (long acting)	1	Liver	7 days (52 ppb)	14 days	Healthy	NS	[154] 2000
					Kidney: 42 ppb				Kidney	14 days (65 ppb)	>14 days			
					Muscle: 45 ppb;				Muscle	7 days (49 ppb)	14 days			
					Fat: 45 ppb				Fat	7 days (88 ppb)	14 days			
									Inj. Site	14 days (59 pb)	>14 days			
Oxytetra-cycline	Sheep; NS; 16 months; *n* = 2 study; *n* = 1/ time pt	US Tol: 6000 ppb (liver); 12,000 ppb (kidney, fat); 2000 ppb (muscle). EMA established MRL for all food producing species: 300 ppb (liver); 600 ppb (kidney); 100 ppb (muscle).	LC-MS	Liver Oxy: 15.3 ppb	Liver Oxy: 50 ppb	IM	10 mg/kg Oxytetracycline daily	5	Liver	Oxy: 2 days (272.8 ppb)	Oxy: >2 days	Healthy	NS	[155] 2008
				Liver 4-Epi ^†^: 16.6 ppb	Liver 4-Epi ^†^: 50 ppb					4-Epi ^†^: 4 h (217.8 ppb)	4-Epi ^†^: 2 days			
				Kidney Oxy: 15.7 ppb	Kidney Oxy: 50 ppb				Kidney	Oxy: 2 days (1342.4 ppb)	Oxy: >2 days			
				Kidney 4-Epi ^†^: 17.5 ppb	Kidney 4-Epi ^†^: 50 ppb					4-Epi ^†^: 2 days (55 ppb)	4-Epi ^†^: >2 days			
				Muscle Oxy: 12.4 ppb	Muscle Oxy: 50 ppb				Muscle	Oxy: 2 days (73.6 ppb)	Oxy: >2 days			
4-epi-Oxytetra-cycline				Muscle 4-Epi ^†^: 13.9 ppb	Muscle 4-Epi ^†^: 30 ppb					4-Epi ^†^: 4 h (34.2 ppb)	4-Epi ^†^: 2 days			
				Fat Oxy: 12.4 ppb	Fat Oxy: 50 ppb				Fat	Oxy: 4 h (3610.7 ppb)	Oxy: 2 days			
				Fat 4-Epi ^†^: 14.1 ppb	Fat 4-Epi ^†^: 30 ppb					4-Epi ^†^: <LOQ @ 4 h	4-Epi ^†^: 4 h			
				Inj. Site Oxy: 12.4 ppb	Inj. Site Oxy: 30 ppb				Inj. Site	Oxy: 2 days (763.2 ppb)	Oxy: >2 days			
				Inj. Site 4-Epi ^†^: 13.9 ppb	Inj. Site 4-Epi ^†^: 30 ppb					4-Epi†: 2 days (34.5 ppb)	4-Epi ^†^: >2 days			
Oxytetra-cycline	Sheep; Chios; 3 years; *n* = 20	US Tol: Not established. EMA established MRL for all food producing species: 100 ppb (milk).	LC-MS	NS	20 ppb	IM	10 mg/kg Oxytetracycline daily	5	Milk	7 days (33.2 ppb)	8 days	Healthy	Lactating; Milked 2×/day	[156] 2008
Oxytetra-cycline	Sheep; Comisana; adult; *n* = 8	US Tol: Not established. EMA established MRL for all food producing species: 100 ppb (milk).	HPLC	NS	NS	IM	20 mg/kg Oxytetracycline (long acting)	1	Milk	7.5 days (50 ppb)	8 days	Healthy	Lactating; Milked 2×/day	[157] 2000
Oxytetra-cycline	Sheep; desert; 9–12 months; *n* = 12/ study; *n* = 4/time pt	US Tol: 6000 ppb (liver);12,000 ppb (kidney, fat); 2000 ppb (muscle). EMA establsihed MRL for all food producing species: 300 ppb (liver); 600 ppb (kidney); 100 ppb (muscle).	Bioassay	NS	NS	IM	5000 mg/kg Oxytetracycline (long acting) daily	5	Liver	10 days (1.51 ppm)	>10 days	NS	NS	[158] 2007
									Kidney	10 days (6.7 ppm)	>10 days			
									Muscle	10 days (70.87 ppm)	>10 days			
									Inj. Site	10 days (1227.7 ppm)	>10 days			
Oxytetra-cycline	Sheep; Chios; 16 months; *n* = 30 study; *n* = 5/time pt	US Tol: 6000 ppb (liver); 12,000 ppb (kidney, fat); 2000 ppb (muscle). EMA established MRL for all food producing species: 300 ppb (liver); 600 ppb (kidney); 100 ppb (muscle).	LC-MS	NS	Liver: 50 ppb	IM	10 mg/kg Oxytetracycline daily	5	Liver	Oxy: 6 days (0.05 ppm)	Oxy: 9 days	Healthy	NS	[146] 2009
										4-Epi ^†^: 2 days (0.05 ppm)	4-Epi ^†^: 4 days			
					Kidney: 50 ppb				Kidney	Oxy: 9 days (0.08 ppm)	Oxy: 12 days			
										4-Epi ^†^: 4 days (0.05 ppm)	4-Epi ^†^: 6 days			
					Muscle: 30 ppb				Muscle	Oxy: 4 days (0.04 ppm)	Oxy: 6 days			
4-epi-Oxytetra-cycline										4-Epi ^†^: 2 days (0.04 ppm)	4-Epi ^†^: 4 days			
					Fat: 30 ppb				Fat	Oxy: 0 days (2.7 ppm)	Oxy: 2 days			
										4-Epi ^†^:<LOQ @ 0 days	4-Epi ^†^: 0 days			
					Inj. Site: 30 ppb				Inj. Site	Oxy: 9 days (0.04 ppm)	Oxy: 12 days			
										4-Epi ^†^: 2 days (0.062 ppm)	4-Epi ^†^: 4 days			
Oxytetra-cycline	Goat; Saanen; adult; *n* = 8	US Tol: Not established. EMA established MRL for all food producing species: 100 ppb (milk).	HPLC	NS	NS	IM	20 mg/kg Oxytetracycline (long acting)	1	Milk	7.5 days (60 ppb)	8 days	Healthy	Lactating; Milked 2×/day	[157] 2000
Oxytetra-cycline	Goat; mixed breed; NS; *n* = 32 Mixed breed; adult; *n* = 10	US Tol: 6000 ppb (liver); 12,000 ppb (kidney, fat); 2000 ppb (muscle); Not approved (milk). EMA established MRL for all food producing species: 300 ppb (liver); 600 ppb (kidney); 100 ppb (muscle); 100 ppb (milk).	Bioassay	0.1 ppm	NS	IM	20 mg/kg Oxytetracycline (long acting)	1	Liver	7 days (385 ppb)	14 days	Healthy	Lactating; Milked 2×/day	[145] 2002
									Kidney	7 days (376 ppb)	14 days			
									Muscle	7 days (246 ppb)	14 days			
									Fat	96 h (236 ppb)	7 days			
									Inj. Site	14 days (1129 ppb)	>14 days			
			HPLC	0.15 ppm		IM	20 mg/kg Oxytetracycline (long acting)	1	Milk	178 h (0.03 ppm)	>178 h	Healthy	Lactating; Milked 2×/day	
						SC	20 mg/kg Oxytetracycline (long acting)	1	Milk	178 h (0.05 ppm)	>178 h			
Oxytetra-cycline	Goat; Saanen; adult; *n* = 8	US Tol: Not established. EMA established MRL for all food producing species: 100 ppb (milk).	Bioassay	0.25 ppm	NS	IMM	426 mg Oxytetracycline per half daily	3	Milk	5 h (0.50 ppm ^§^)	>5 h	Healthy	Lactating; Milked 2×/day	[53] 1984
Oxytetra-cycline	Goat; NS; adult; NS	US Tol: Not established. EMA established MRL for all food producing species: 100 ppb (milk).	HPLC	NS	NS	IM	15 mg/kg Oxytetracycline daily	4	Milk	100 h (0.46 ppm)	>100 h	Healthy	Lactating	[159] 1994
Oxytetra-cycline	Goat; Canary Island; adult; *n* = 5	US Tol: Not established. EMA established MRL for all food producing species: 100 ppb (milk).	HPLC	NS	NS	IM	15 mg/kg Oxytetracycline	4	Milk	96 h (0.46 ppm)	>96 h	Healthy	Lactating; Milked 2×/day	[160] 1995
Oxytetra-cycline	Goat; Saanen; adult; *n* = 8	US Tol: Not established. EMA established MRL for all food producing species: 100 ppb (milk).	LC-MS	15 ppb	50 ppb	IM	20 mg/kg Oxytetracycline	1	Milk	180 h (60 ppb)	8 days	Healthy	Lactating; Milked 2×/day	[161] 2002
Oxytetra-cycline	Goat; Nubian, Alpine & LaMancha; adult; *n* = 15	US Tol: Not established. EMA established MRL for all food producing species: 100 ppb (milk).	HPLC	NS	NS	IM	17.6 mg/kg Oxytetracycline at 48 h interval	2	Milk	96 h (87 ppb)	>96 h	Healthy	Mid-lactation; Milked 2×/day	[144] 2015
Oxytetra-cycline	Goat; Murciano-Granadina; adult; *n* = 5	US Tol: Not established. EMA established MRL for all food producing species: 100 ppb (milk).	HPLC	NS	NS	IV	20 mg/kg Oxytetracycline chlorhydrate	1	Milk	2 days (0.25 ppm ^§^)	>2 days	Healthy	Lactating; Milked 1×/day	[162] 2001
						IM	20 mg/kg Oxytetracycline chlorhydrate	1	Milk	3 days (0.36 ppm ^§^)	>3 days			
						IM	20 mg/kg Oxytetracycline dehydrate (Long Acting)	1	Milk	3 days (0.27 ppm ^§^)	>3 days			
Oxytetra-cycline	Goat; NS; 2–7 years; *n* = 10	US Tol: Not established. EMA established MRL for all food producing species: 100 ppb (milk).	Bioassay	NS	NS	IMM	426 mg Oxytetracycline hydrochloride per half at 12 h intervals	3	Milk	96 h (0.02 ppm ^§^)	108 h	Healthy	Early & mid-lactation; Milked 2×/day; 1 tube/ mammary gland	[60] 1984
Tetra-cycline	Sheep; Awassi; Adult; *n* = 2	US Tol: Not established. EMA established MRL for all food producing species: 100 ppb (milk).	Bioassay	NS	NS	IM	20 mg/kg Tetracycline	1	Milk	48 h (0.08 ppm ^§^)	>2 days	Healthy	Lactating; Milked 2×/day	[29] 1974
										48 h (0.04 ppm ^§^)	>2 days	Diseased- mastitis		
			Radio-activity	NS	NS	IM	20 mg/kg Tetracycline (radiolabeled)	1	Milk	48 h (0.14 ppm ^§^)	>2 days	Healthy		
										48 h (0.2 ppm ^§^)	>2 days	Diseased- mastitis		
Tetra-cycline	Sheep; Awassi; adult; *n* = 4	US Tol: Not established. EMA established MRL for all food producing species: 100 ppb (milk).	Bioassay	NS	NS	IV	20 mg/kg Tetracycline hydrochloride (radiolabeled), then 5 mg/kg for 2 doses at 90 min interval	1	Milk	60 h (0.70 ppm ^§^)	4 days	Healthy	Lactating; Milked 2×/day	[24] 1973
			Radio-activity	NS	NS					60 h (0.12 ppm ^§^)	4 days			
Tetra-cycline	Goat; Canary Island; adult; *n* = 5	US Tol: Not established. EMA established MRL for all food producing species: 100 ppb (milk).	HPLC	NS	NS	IM	15 mg/kg Tetracycline	4	Milk	96 h (0.91 ppm)	>96 h	Healthy	Lactating; Milked 2×/day	[160] 1995
Tetra-cycline	Goat; NS; adult; NS	US Tol: Not established. EMA established MRL for all food producing species: 100 ppb (milk).	HPLC	NS	NS	IM	15 mg/kg Tetracycline daily	4	Milk	100 h (0.91 ppm)	>100 h	Healthy	Lactating	[159] 1994

^§^ Data points manually extracted use scanning software (Webplot digitizer or UnScanIt 7.0). # = number. ^†^ 4-epi-Oxytetracycline Metabolite. * Projected time for which residues could still be detected based on study protocol for sample collection time points and sample concentration results. Authors caution readers to critically evaluate these publications to estimate when full residue depletion might occur. Abbreviations: 2×/day: twice daily. LOD: Limit of detection. LOQ: Limit of quantification. EMA: European Medicines Agency. MRL: Maximum residue limit. ND: Not detected. NS: Not specified. Routes of Administration: IMM = intramammary, IM = intramuscular, IV = intravenous, PO = per os, POMF = per os as medicated feed, SC= subcutaneous. Units: s = seconds, min = minutes, h = hours, ppb = parts per billion, ppm = parts per million.

## Data Availability

The original contributions presented in this review are included in the main manuscript and included tables. Further inquiries can be directed to the corresponding author.

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
