# Peer review of "Antibacterial Drug Residues in Small Ruminant Edible Tissues and Milk: A Literature Review of Commonly Used Medications in Small Ruminants"

_animals, 2022, doi:10.3390/ani12192607_

Round 1

Reviewer 1 Report

Ms. Ref. No.: Animals-2022,12

Title: Drug residues in small ruminant edible tissues and milk: a literature review of commonly used medications in small ruminants

Journal: Animals

REFEREE'S COMMENTS

The work is an extensive review that report a summary of extracted data from the published literature about antimicrobial drug residue in milk and other edible tissues after administration to sheep and goats. An exhaustive study of the revised works is made and data of great interest are reported. The information collected can be useful to different regulatory bodies and/or veterinarians to make decisions on the rational use of these drugs in sheep and goats. However, some revision needs to be made for a suitable presentation of the manuscript in Animals.

General comment

The work reports a large amount of data obtained from published works reporting antimicrobial residues in milk/tissues after administration by different routes to sheep and goat. Data are arranged by groups of antibacterial drugs and reported in tables with a brief introductory paragraph before each table. The main criticism of the review is that the collected data allows a much broader analysis, discussion and conclusions than what was reported.

The following specific comments must be considered to prepare a revised version of this manuscript.

Page 1:

Title: “Drug residues in small ruminant edible tissues and milk: a literature review of commonly used medications in small ruminants”

·       According to the title, it is expected to find some data on levels of antibiotic residues in milk and other tissues, but these are not reported in this review.  Authors are recommended to reformulate the title in this sense.

·       It would be of great value if the authors include some value for antimicrobial residue levels in the table, at least some of the decreasing terminal phase. This is relevant, especially considering that to determine the withdrawal period, the level of quantified residues must be taken into account and not only the period detecting residues. As authors mentioned in the manuscript, not all reported analytical techniques have the same sensitivity. For a technique that is not very sensitive, the fact that no residues are detected at a certain time does not mean that there are no residues. On the contrary, it may happen that residues are detected for some time but at very low levels.

·       Since only studies on antimicrobials were reviewed and other important medications prescribed in ruminants, such as antiparasitic drugs, are not included, the wording should be more specific. It is recommended to replace the term "drug residues" with "antimicrobial residues" in the abstracts, keywords and in the title.

Page 2:

Line 57: “…are considered minor species by the European Medicines Agency (EMEA)”

Page 3:

Line 130:Tolerances or maximum residue limits are presented for FDA-approvals and EMEA-approvals, respectively.”

·       Currently EMEA is named EMA, please correct it.

Page 4:

Line 164: “Amphenicols, some penicillins (antipseudomonal, aminopenicillins with and without beta-lactamase inhibitors) as highly important antimicrobials for human health by the WHO. Some penicillins (amidinopenicillins, anti-staphylococcal, narrow spectrum), sulfonamides, tetracyclines are classified as highly important antimicrobials for human health by the WHO.

·       These two sentences are referring to the same, please to simplify into one.

Page 4:

Line 189: “However, the EMEA has approved…

·       Currently EMEA is named EMA, please correct it.

Page 4:

Line 170: “Aminoglycosides (amikacin, apramycin, dihydrostreptomycin, gentamicin, tobramycin, neomycin, spectinomycin) are…”

·       Spectinomycin is not an aminoglycoside but an aminocyclitol, rewrite the paragraph considering this.

Page 4:

Line 218: “3.3. Beta-Lactams/Penicillins

Beta-lactams (amoxicillin, ampicillin, cloxacillin, dicloxacillin, nafcillin), including penicillins (penicillin G procaine, penicillin G benzathine), are bactericidal…”

Since the group of beta-lactams does not include only penicillins, it is recommended to rewrite this sentence (following the format of point 3.4.) for a better understanding.

Page 39:

Line 336: “Marcolies (erythromycin..”

·       Spell macrolides correctly.

Page 47:

Line 375: “Sulfonamides (…) are bacteriostatic medications that complete with…”

·       It would be more appropriate to say "Sulfonamides (…) are bacteriostatic antimicrobial drugs that compete with …”

Page 47:

Line 383: “and sulfacetamide) are found in in the milk in lower concentrations…”

·       Delete one “in”

Page 70:

Line 801, Reference 162: Rule, R.M., L.; Serrano, J.M.; Garcia Roman, A.; Moyano, R.; Garcia, J. Pharmacokinetics and residues in milk of 801 oxytetracyclines administered parenterally to dairy goats. Aust. Vet. J. 2001, 79, 492-496.

·       Should say: Rule, R.M., Moreno, L.; Serrano, J.M.; Garcia Roman, A.; Moyano, R.; Garcia, J. Pharmacokinetics and residues in milk of 801 oxytetracyclines administered parenterally to dairy goats. Aust. Vet. J. 2001, 79, 492-496.

Tables

Table headlines: Ej.“Table 1: Aminoglycoside residues in milk or edible tissue samples from sheep or goats following treatment”

·       Unless the residue values in milk or edible tissue values are included in the revised version of the manuscript, all table headlines should be changed, since data on antimicrobial residue levels after treatment are not reported in tables.

First row, when says “route”

·       Replaces by route of administration

Table 1 and 2.

·       The meaning of the abbreviations was not included in these tables. Most are self-explanatory, others are not. What does POMW mean?.

Table 5

Row of Reference 93. Values of 0.05 and 0.04 ppm are reported for LOD and LOQ, respectively.

·       Please check these data, since LOD can not be higher than LOQ.

Reviewer 2 Report

The authors should consider the followings:

1.          The authors should state the age and subtypes of goat and/or sheep in their table content, since the ages (juvenile goat etc) and the animal subtypes would have a different ADME profiles.

2.          The authors should clearly specify whether they were summarizing the parent and/or metabolite(s) in their table content.

3.          The authors should indicate the number of animal(s) used in the respective studies.

4.          The authors may provide a global perspective of the relevant Food and Agriculture regulation (such as those in Australia, New Zealand, China and India), regarding the Food Animal Residue.

5.          The authors may illustrate the publication screening results as a flowchart.

6.          Since the topic of the review is for edible tissues and milk, the authors should consider whether to enlist the diseased goat/ sheep in the review tables.

7.          The authors may add a column for the year of publication to those tables; this may be helpful to overview the relevance measurement uncertainty, since the authors cover the year of publication back to 1920s.

8.          In small ruminants, the authors may give rationale(s) of only choosing sheep and goats, in the introduction and/or method selection.

9.          The authors may split the table, according to the health status of the animal, for example, the authors should group the "disease", "NS" status of Health status to another tables.

10.      The authors may use or compile supplementary tables (Excel .xls for example) for audience to track the tabulation of results mentioned in the main articles.

11.      The authors may give rationales on why the authors confine the “drug residues” to only antimicrobial residues, rather than the other medication in this current review.
